# Phosphate limitation intensifies negative effects of ocean acidification on globally important nitrogen fixing cyanobacterium

Futing Zhang [1,4,5], Zuozhu Wen[1,5], Shanlin Wang[1], Weiyi Tang[2], Ya-Wei Luo [1],
Sven A. Kranz [3], Haizheng Hong [1] ✉ & Dalin Shi [1] ✉

Growth of the prominent nitrogen-fixing cyanobacterium *Trichodesmium* is often limited by phosphorus availability in the ocean. How nitrogen fixation by phosphorus-limited *Trichodesmium* may respond to ocean acidification remains poorly understood. Here, we use phosphate-limited chemostat experiments to show that acidification enhanced phosphorus demands and decreased phosphorus-specific nitrogen fixation rates in *Trichodesmium*. The increased phosphorus requirements were attributed primarily to elevated cellular polyphosphate contents, likely for maintaining cytosolic pH homeostasis in response to acidification. Alongside the accumulation of polyphosphate, decreased NADP(H):NAD(H) ratios and impaired chlorophyll synthesis and energy production were observed under acidified conditions. Consequently, the negative effects of acidification were amplified compared to those demonstrated previously under phosphorus sufficiency. Estimating the potential implications of this finding, using outputs from the Community Earth System Model, predicts that acidification and dissolved inorganic and organic phosphorus stress could synergistically cause an appreciable decrease in global *Trichodesmium* nitrogen fixation by 2100.

In vast areas of the oceans, photosynthetic carbon fixation by marine primary producers is constrained by the deficiency of nitrogen (N)[1]. The bloom-forming cyanobacteria *Trichodesmium* spp. are estimated to contribute up to 50% of marine $N_2$ fixation that supports primary production in the low-nutrient open ocean ecosystems[2]. The concentration of carbon dioxide ($CO_2$) in ambient seawater, however, can potentially limit the growth of *Trichodesmium*, owing to the exceptionally low catalytic efficiency of their ribulose-1,5-bisphosphate carboxylase/oxygenase (Rubisco)[3], the key enzyme that catalyzes $CO_2$ fixation. *Trichodesmium* thus relies on the energetically demanding carbon concentrating mechanism (CCM) to elevate intracellular $CO_2$ concentration at the site of carbon fixation by Rubisco[4]. Therefore, an increase in partial pressure of seawater $CO_2$ ($pCO_2$), as a result of ocean

acidification (OA)[5], has been hypothesized to benefit *Trichodesmium* by down-regulating the CCM and thereby allowing saved energy to be reallocated to and used by other cellular processes such as $N_2$ fixation[4]. Recent studies have shown that the positive effects of increased $pCO_2$ on *Trichodesmium erythraeum* IMS101 (*T. erythraeum*) are, however, relatively limited and can be offset by the deleterious effects of the concurrently decreased seawater pH[6,7]. A negative OA effect on growth and $N_2$ fixation has been shown, both in the laboratory and in the field, for this diazotrophic organism, especially under limitation by the micronutrient iron (Fe)[6,8].

Aside from Fe, the growth of *Trichodesmium* is often limited by the essential macronutrient phosphorus (P)[9], the supply of which (and other nutrients including N) to oligotrophic surface waters is expected

[1]State Key Laboratory of Marine Environmental Science, Xiamen University, Xiamen, Fujian, PR China. [2]Department of Geosciences, Princeton University, Princeton, NJ 08544, USA. [3]Department of Earth, Ocean and Atmospheric Science, Florida State University, Tallahassee, FL 32306, USA. [4]Present address: The Freddy and Nadine Herrmann Institute of Earth Sciences, Hebrew University of Jerusalem, Jerusalem, Israel. [5]These authors contributed equally: Futing Zhang, Zuozhu Wen. ✉e-mail: honghz@xmu.edu.cn; dshi@xmu.edu.cn

to decrease in the future ocean[10]. The effects of OA on *Trichodesmium* are likely modulated by P availability due to its roles in many central cellular functions such as energy generation, genetic information, and cellular structure. Paradoxically, some studies have shown that P limitation does not modulate the effect of acidification on growth and N$_2$ fixation of *T. erythraeum*[11,12], whereas others have reported considerable enhancement of the acidification impact under P limitation on the same cyanobacteria cultured under similar conditions[13]. In addition, these studies were all conducted with *T. erythraeum* grown in an artificial seawater medium under non-steady-state P-limited conditions using dilute batch cultures[11,13].

In this study, we use phosphate (PO$_4^{3-}$)-limited chemostats to simulate in situ P limitation using a natural seawater medium (i.e., Aquil-tricho)[6]. This different culturing approach enables us to examine the response of *T. erythraeum* to acidification and the underpinning mechanisms under steady-state P limitation, mimicking continued but limiting P supply to the upper ocean. The laboratory studies are further corroborated by ship-board incubation experiments with naturally occurring *Trichodesmium* populations in the northern South China Sea (NSCS). We find that OA leads to increased P requirements and decreased P-specific N$_2$ fixation rates in P-limited *Trichodesmium*, and the OA-induced adverse effects are more marked under P limitation than under P sufficiency. By using an offline mode of the Community Earth System Model (CESM), we estimate that interactions between P limitation and acidification will probably result in an appreciable decrease in *Trichodesmium* N$_2$ fixation in the future ocean.

## Results and discussion

### Effects of OA on *T. erythraeum* under steady-state P limitation

*T. erythraeum*, which had been semicontinuously cultured at 0.5 μM PO$_4^{3-}$ under ambient or acidified (400 or 750 μatm, Supplementary Table 1) condition for more than one year, was subsequently grown for about one month in PO$_4^{3-}$-limited chemostats. The steady-state soluble reactive phosphorus (SRP) concentrations in the chemostats were maintained at 12.4–16.3 nM (Supplementary Fig. 1), which is within the range of dissolved inorganic P concentration typically observed in surface waters of oligotrophic oceans which *Trichodesmium* inhabits[14–16]. Under ambient conditions, the ratio of cellular particulate organic nitrogen to phosphorus (PON:POP) was 48:1 (Supplementary Table 2), which was much higher than that under P-replete conditions (20:1, as observed in a previous study[8]) and was also close to those of *Trichodesmium* spp. in P-limited North Atlantic[15,17]. In addition, the activity of alkaline phosphatase, a molecular diagnostic of P stress in *Trichodesmium*[18], observed when the cultures reached steady-state

conditions was significantly higher than that under P-replete conditions (Supplementary Fig. 2a). These data suggest that *T. erythraeum* in our chemostats experienced P limitation.

At a given growth rate of 0.2 day$^{-1}$ (Supplementary Note 1), acidification resulted in a significant decrease of 30% ($p = 0.024$, two-tailed paired Student's *t*-test) in the cell number-normalized N$_2$ fixation rate of *T. erythraeum* under P-limited conditions (Fig. 1a), as compared to 13% under P-replete conditions observed previously[8]. The C fixation rate also decreased more markedly under P limitation (17%, Fig. 1a) than under P sufficiency (8%, as observed previously[8]). In accordance with the decreased rates, cellular particulate organic nitrogen and carbon (PON and POC) contents decreased by 18% and 22%, respectively, at high *p*CO$_2$/low pH (Fig. 1b). Intriguingly, under acidified conditions POP of the P-limited *T. erythraeum* increased by 28% ($p < 0.001$, two-tailed paired Student's *t*-test) (Fig. 1b), despite a slight decrease in cell size (Supplementary Table 2). As a result, the PON:POP and POC:POP ratios decreased to 31:1 and 202:1, respectively, which were 35% and 39% ($p < 0.001$, two-tailed paired Student's *t*-test) lower than those under ambient conditions (48:1 and 332:1, respectively, Supplementary Table 2). Normalizing N$_2$ and C fixation rates to the cellular PON and POC, respectively, showed that under acidified conditions the N-specific N$_2$ fixation rate decreased by 16% ($p = 0.068$, two-tailed paired Student's *t*-test) and the C-specific C fixation rate was generally not affected ($p = 0.327$, two-tailed paired Student's *t*-test) (Supplementary Table 2). Because both the PON:POP and POC:POP ratios decreased significantly under acidified conditions, P-limited *T. erythraeum* may fix less N and C per unit of P in response to acidification. This is also supported by the reduction of N$_2$ and C fixation rates normalized to cellular POP (45 % lower, $p = 0.003$; and 36% lower, $p = 0.007$, respectively, two-tailed paired Student's *t*-test) (Supplementary Table 2).

To systematically explore how acidification led to enhanced POP under P limitation, we conducted the transcriptomic analysis with the PO$_4^{3-}$-limited *T. erythraeum* grown under both ambient and acidified conditions. Compared to ambient conditions, acidification showed little influence on the transcription of a suit of high-affinity PO$_4^{3-}$ transporters (*pstSCAB*) (Supplementary Fig. 2b). Aside from PO$_4^{3-}$, *Trichodesmium* spp. are able to utilize phosphite and dissolved organic phosphorus (DOP), including the monophosphate esters (C–O–P bond) and phosphonates (C–P bond), to cope with P limitation[19,20]. In our culture media, DOP was not the added P source and was estimated to support only a few percent of POP at the most (Supplementary Table 3). We found that gene transcriptions of alkaline phosphatases (e.g., *phoX2* and *phoA*)[21] were not significantly affected while those

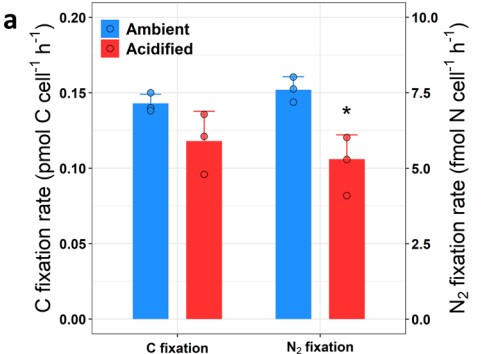
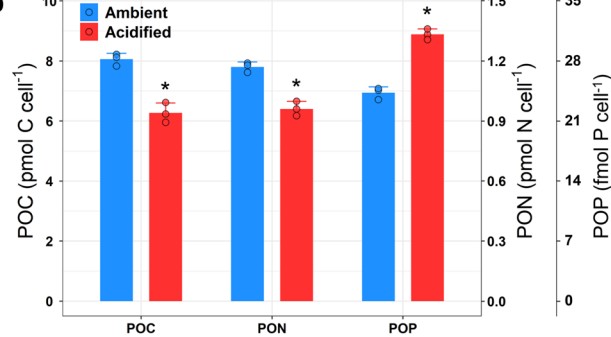
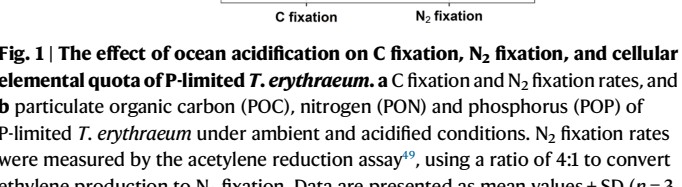

**Fig. 1 | The effect of ocean acidification on C fixation, N$_2$ fixation, and cellular elemental quota of P-limited *T. erythraeum*. a** C fixation and N$_2$ fixation rates, and **b** particulate organic carbon (POC), nitrogen (PON) and phosphorus (POP) of P-limited *T. erythraeum* under ambient and acidified conditions. N$_2$ fixation rates were measured by the acetylene reduction assay[49], using a ratio of 4:1 to convert ethylene production to N$_2$ fixation. Data are presented as mean values + SD ($n = 3$

biologically independent samples), and dots are corresponding data points of the replicates. Asterisks denote significant changes in N$_2$ fixation rates ($p = 0.024$) and POC ($p = 0.001$), PON ($p = 0.001$), and POP ($p < 0.001$) under acidified conditions compared with ambient conditions (two-tailed paired Student's *t*-test). Source data are provided as a Source Data file.

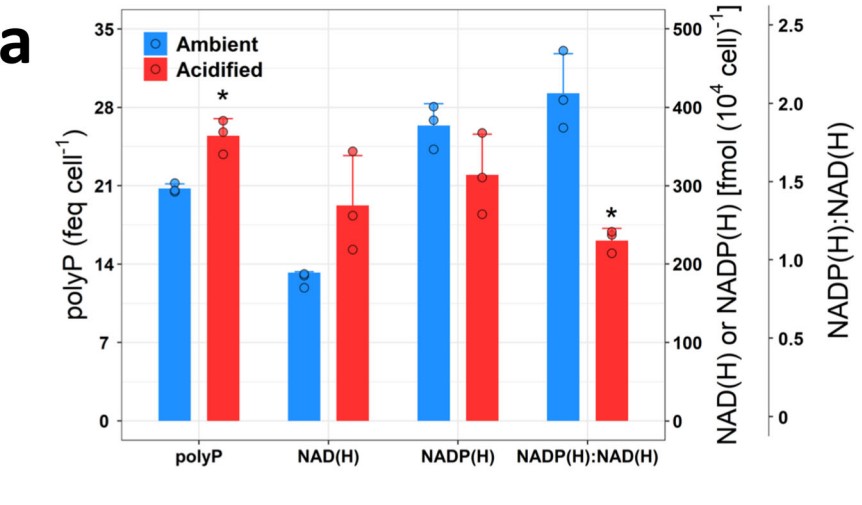

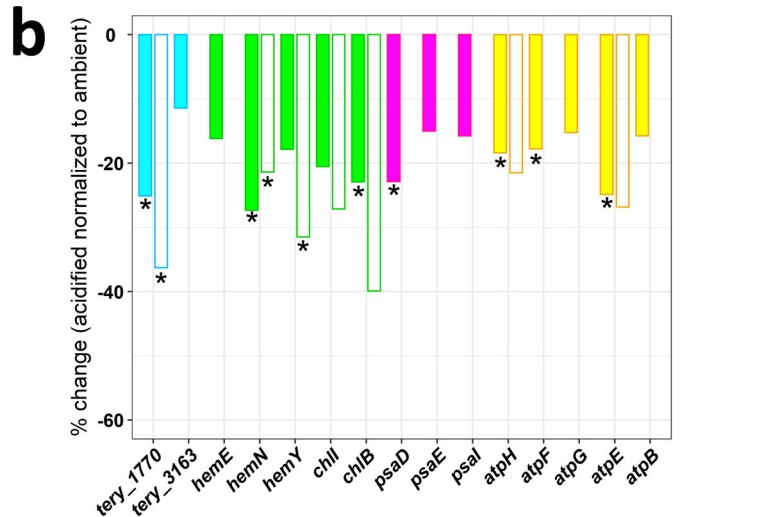

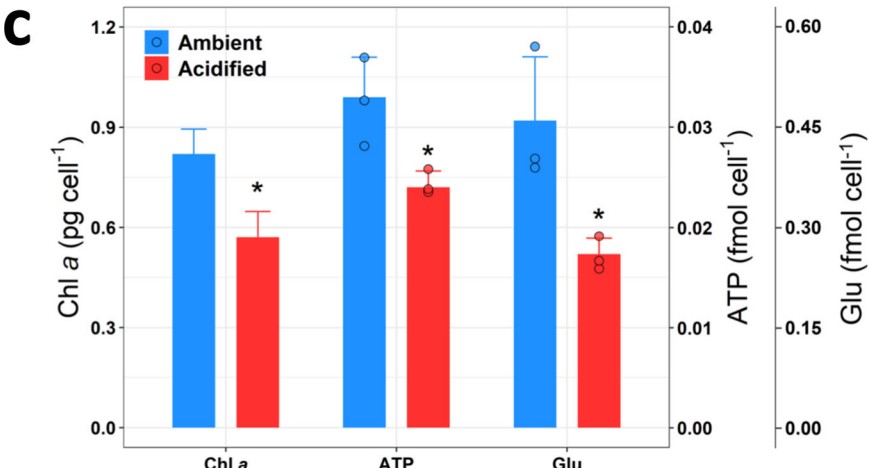

involved in phosphonate (and/or phosphite) utilization (i.e., *phnD1/ptxA*, *phnC1/ptxB*, *phnE1/ptxC*, *ptxD*, and *Tery_2885*) were markedly upregulated by OA (Supplementary Fig. 2b). Overall, these gene transcription results suggest an increased P requirement in P-limited *T. erythraeum* under acidified conditions.

Using a fluorometric quantification method[22,23], we observed that acidification resulted in a significant enhancement (23%, *p* < 0.007, two-tailed paired Student's *t*-test) of the relative polyphosphate (polyP) concentration (expressed as femto-equivalents of the standard

per cell; see the "Methods" section) in the P-limited *T. erythraeum* (Fig. 2a). P content in lipid, DNA, and RNA (accounting for ~24% of POP), however, was not largely affected by acidification (Supplementary Table 2). Therefore, in view of being a substantial portion of POP[23,24], polyP should be mostly responsible for the enhanced POP under acidified conditions (Fig. 1b). A ubiquitous polymer of $PO_4^{3-}$ residues, polyP is known as a luxury P storage molecule in phytoplankton[23,24]. In addition, polyP also has diverse physiological functions in microorganisms, such as maintaining intracellular pH

**Fig. 2 | The effect of ocean acidification on cellular metabolite, pigment, and gene transcription of P-limited *T. erythraeum*. a** PolyP (femto-equivalents of the standard per cell), NAD(H) and NADP(H) concentrations, and NAD(H):NADP(H) ratios of P-limited *T. erythraeum* under ambient and acidified conditions. **b** Percentage change (acidified normalized to ambient condition) of gene transcription of NAD kinase (cyan) and proteins involved in Chl*a* synthesis (green), PSI (purple), and ATP synthesis (yellow). Solid bars denote transcriptomic analysis data, and open bars denote RT-qPCR analysis data (Supplementary Tables 4 and 5). **c** cellular Chl*a*, ATP, and Glu concentrations of P-limited *T. erythraeum* under ambient and acidified conditions. In **a**, **c**, data are presented as mean values + SD ($n = 3$ biologically independent samples, except for Chl*a* $n = 14$ and 15 under

ambient and acidified conditions, respectively), and dots are corresponding data points of the replicates. Asterisks denote significant changes in polyP ($p = 0.007$), NAD(H):NADP(H) ratios ($p = 0.003$), Chl*a* ($p < 0.001$), ATP ($p = 0.036$), and Glu ($p = 0.031$) under acidified conditions compared with ambient conditions (two-tailed paired Student's *t*-test). In **b**, data are mean values of three biological replicates ($n = 3$ biologically independent samples), and asterisks denote significant changes ($p < 0.05$) in gene transcription in response to acidification (for the transcriptomic data analysis, differential expression was analyzed using the DESeq2 R package[54] and the *p*-values were adjusted using the Benjamini and Hochberg's approach; for RT-qPCR data analysis, two-tailed paired Student's *t*-test). Source data are provided as a Source Data file.

homeostasis thanks to its ability to act as a proton ($H^+$ ion) buffer[25]. For example, a pH-homeostatic function for polyP has been demonstrated in the unicellular alga *Dunaliella salina*[26], and the reduced pH in the growth medium was found to result in a significant intracellular accumulation of polyP in the yeast *Candida humicola*[27]. Recently Hong et al.[6] have reported that OA can lead to a decrease in the cytosolic pH of *T. erythraeum*, thereby affecting intracellular pH homeostasis and accordingly increasing biochemical and energetic costs for coping with it. In marine phytoplankton, the cellular mechanisms for cytosolic pH homeostasis generally include $H^+$ buffering, metabolic $H^+$ consumption and production, and transmembrane $H^+$ transport[28,29]. In this study as well as a previous one[6], the gene expression of several plasma membrane transporters was increased under acidified conditions (Supplementary Table 4). Therefore, it is plausible that among other mechanisms such as up-regulating transmembrane $H^+$ transport machinery, the P-limited *T. erythraeum* increased cellular polyP content for maintaining cytosolic pH homeostasis in response to acidification.

Although cellular polyP concentration increased under acidified conditions, the transcription of polyP kinase and phosphatases, enzymes that catalyze polyP synthesis and hydrolysis, respectively, was not substantially affected by acidification (Supplementary Table 4). As demonstrated by both transcriptomic and RT-qPCR analysis, the transcription of *tery_1770* and *tery_3163*, genes encoding nicotinamide adenine dinucleotide (NAD) kinase (i.e., *ppnk*), was down-regulated by acidification (Fig. 2b and Supplementary Tables 4 and 5). NAD kinase catalyzes the phosphorylation of NAD to the formation of nicotinamide adenine dinucleotide phosphate (NADP) in the presence of phosphoryl donors such as polyP and ATP[30]. Although the physiological functions of NAD kinase in cyanobacteria remain poorly understood[31] and its phosphoryl donor(s) in *T. erythraeum* is(are) unknown, the down-regulation of NAD kinase under acidified conditions may have helped sustain higher polyP content in the cells, which could play an important role in cytosolic pH homeostasis, as discussed above.

NAD kinase is one of the key enzymes that regulate the cellular NADP(H) and NAD(H) levels and the NADP(H):NAD(H) ratios[30]. In accordance with the down-regulation of NAD kinase gene transcription, acidification was found to cause an increase in NAD(H) and a decrease in NADP(H) concentrations and hence a decrease in the NADP(H):NAD(H) ratio ($p = 0.003$, two-tailed paired Student's *t*-test) (Fig. 2a). It has been shown that the NADP(H):NAD(H) ratio plays an important role in regulating plant physiology, as the functions of NAD(H) and NADP(H) are important and distinct[30,32]. For instance, in NAD kinase-deficient *Arabidopsis*, the NADP(H):NAD(H) ratio was significantly reduced and the NADPH-dependent pathways such as C fixation and chlorophyll synthesis were hampered[32,33]. Similarly, in our study, along with the decreased NADP(H):NAD(H) ratio under acidified conditions (Fig. 2a), we observed a considerable decrease in C fixation rate (Fig. 1a), as well as a lower cellular chlorophyll *a* (Chl*a*) content (Fig. 2c) and accordingly a significant decrease in the transcription of a series of genes involved in Chl*a* biosynthesis (i.e., *hemN*, *hemY* and *chlB*) (Fig. 2b and Supplementary Tables 4 and 5). The decrease in

cellular Chl*a* concentration was also in line with a reduction in the cellular content of glutamate (Fig. 2c), the substrate for Chl*a* synthesis. In cyanobacteria, it has been suggested that the de novo Chl*a* synthesis reflects the demand of Chl*a* for the photosystem I (PSI)[34], as it contains a much larger number of Chl*a* molecules than the photosystem II (PSII) (i.e., 96 vs. 35 per monomer)[35,36]. Consistent with the reduced cellular Chl*a* content, the transcription of PSI genes (e.g., *psaD*) was decreased in response to acidification (Fig. 2b and Supplementary Table 4), which would have hampered the light-harvesting ability of PSI and consequently ATP production. This is evidenced by a lower cellular ATP content and a decline in the transcription of several ATPase subunit genes (e.g., *atpH*, *atpF*, and *atpE*) at low pH (Fig. 2b, c and Supplementary Tables 4 and 5). Alongside the decreased NADP(H):NAD(H) ratio (Fig. 2a), a shortage of ATP supply should have also been responsible for the decreased C and $N_2$ fixation rates under acidified conditions (Fig. 1a). Moreover, poised with a lower cytosolic pH[6], a reduced ATP pool may cause the nitrogenase complex to allocate a greater fraction of electrons to protons instead of $N_2$[37,38], thereby resulting in a lower nitrogenase efficiency under acidified conditions. This is demonstrated by reduced $N_2$ fixation rates despite elevated transcriptions of nitrogenase genes (e.g., *nifB*, *nifH*, and *nifW*) (Fig. 1a and Supplementary Table 4), as observed previously in both nutrient-replete and Fe-limited *T. erythraeum* grown under low pH[6,8]. In addition, since ATP production was impeded, the energy required for maintaining cytosolic pH homeostasis by plasma membrane transporters would likely become insufficient. Consequently, P-limited *T. erythraeum* would have relied on the up-regulation of cellular polyP to maintain cytosolic pH homeostasis under acidified conditions.

### Effects of OA on natural *Trichodesmium* spp. in NSCS surface seawaters

Previous studies have shown that the effects of OA on $N_2$ fixation of *T. erythraeum* can be influenced by Fe nutritional conditions (Fe-limited vs. Fe-replete)[6,8], and may also depend on their morphologies (large colonies vs. free trichomes)[39,40]. Results from the laboratory culture experiments of the present study further predict that $N_2$ fixation of *Trichodesmium* in natural waters where P is in short supply should decrease with decreasing pH to a larger extent as compared with those from P-sufficient environments. To our knowledge, however, there has been no field study of the effect of acidification on $N_2$ fixation under P-limiting conditions. To test whether our laboratory results can be extended to natural *Trichodesmium* populations, we conducted field experiments with surface seawaters of the northern South China Sea, where SRP concentrations were very low (10−35 nM, Supplementary Table 6). A down-regulation in the expression of the P limitation marker gene *sphX*[21] in response to additions of $PO_4^{3-}$ indicated that the natural *Trichodesmium* populations in surface waters collected in our study region were experiencing P limitation (Supplementary Fig. 3). In all the experiments, *Trichodesmium* spp. accounted for a significant fraction of the diazotroph community (45−98%), according to *nifH* gene abundance (Fig. 3a), and were present as free trichomes except at station OA-5 where *Trichodesmium* formed colonies in a bloom. If not

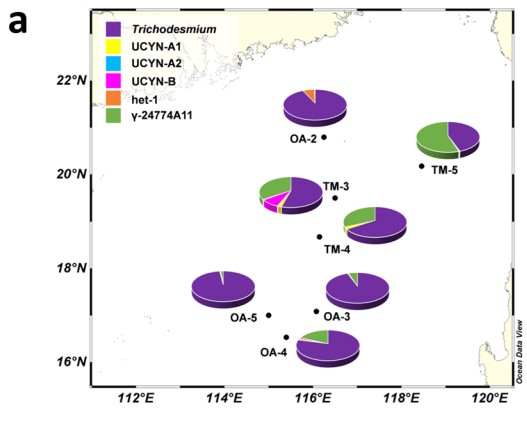
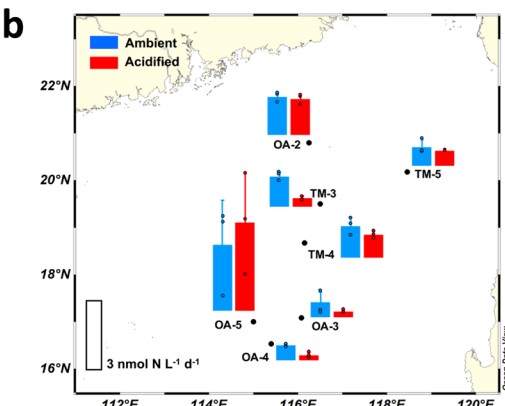
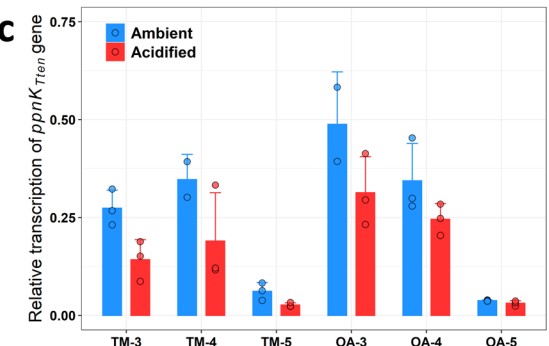
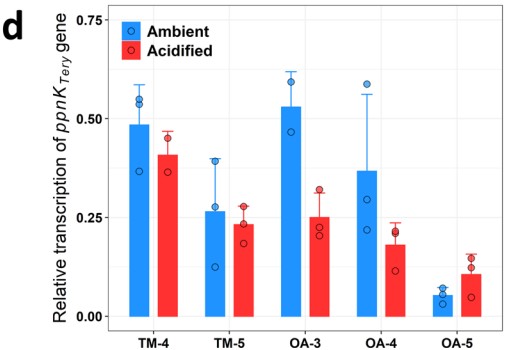

**Fig. 3 | The effect of ocean acidification on natural *Trichodesmium* populations in the northern South China Sea surface seawater. a**, **b** Relative contribution of different diazotrophs (**a**) and $N_2$ fixation rate (**b**) under ambient and acidified conditions of the diazotroph community. **c**, **d** Relative transcription of the NAD kinase gene *ppnk* of two major *Trichodesmium* clades *T. tenue* (**c**) and *T. erythraeum* (**d**) in the diazotroph community under ambient and acidified conditions. In **a**, **b** the maps were produced using Ocean Data View 5. In **b**–**d**, data are presented as mean values + SD ($n = 2$ or 3 biologically independent samples). Although the OA effect in any given experiment was not statistically significant, statistical analysis (one-tailed paired Student's *t*-test) on data from all the experiments except the one at station OA-5, where *Trichodesmium* formed colonies in a bloom, reveals that differences between ambient and acidified conditions were significant for $N_2$ fixation rate (**b**, $p = 0.006$, $n = 18$ for both conditions) and relative transcription of $ppnk_{Tten}$ (**c**, $p = 0.031$, $n = 13$ for ambient conditions and $n = 15$ for acidified conditions) and $ppnk_{Ttery}$ (**d**, $p = 0.008$, $n = 11$ for both conditions). Source data are provided as a Source Data file.

taking into account gamma-proteobacteria whose nitrogenase activity is currently unclear[41], *Trichodesmium* spp. dominated the other surveyed $N_2$ fixers (>83%). As predicted by the laboratory experiments, the $N_2$ fixation rate decreased by 5–73% ($p = 0.006$, one-tailed paired Student's *t*-test) under acidified conditions in all experiments except that at station OA-5 (Fig. 3b). Although natural phytoplankton assemblages in surface waters precluded POP and/or polyP quantification specifically targeting the *Trichodesmium* populations, transcripts of the NAD kinase gene *ppnk* of two major *Trichodesmium* clades (i.e., *T. tenue* and *T. erythraeum*) showed a systematic and significant decline under acidified conditions ($p = 0.031$ and 0.008, respectively, one-tailed paired Student's *t*-test) (Fig. 3c, d). This is consistent with the decreased transcription of the NAD kinase genes observed in our laboratory experiments (Fig. 2b), indicating a down-regulation of NAD kinase expression potentially related to the need for more cellular polyP likely for maintaining cytosolic pH homeostasis.

## Simulated synergistic effects of OA and P limitation on global *Trichodesmium* $N_2$ fixation

The Community Earth System Model (CESM) predicts that the global marine $N_2$ fixation of 155 Tg N yr⁻¹ in 1981–2000 will be reduced by 25 Tg N yr⁻¹ by 2081–2100 under the RCP 8.5 scenario. We estimated the contribution of *Trichodesmium* to $N_2$ fixation in the global ocean following Tang and Cassar[42] and predicted that it will decrease from 60 Tg N yr⁻¹ in 1981–2000 to 47.4 Tg N yr⁻¹ in 2081–2100 (Fig. 4a), a reduction of 12.6 Tg N yr⁻¹. As in most large-scale Earth systems or global ocean models, CESM does not explicitly represent the effects of

acidification on phytoplankton including diazotrophs. To illustrate the potential implications of our findings, using an offline mode of the biogeochemical elemental cycling (BEC) model embedded in the CESM, we extrapolated the acidification effect without considering potential P limitation, which showed an additional decrease of 11.3 Tg N yr⁻¹ in the projected *Trichodesmium* $N_2$ fixation (Fig. 4b). By overlying the synergistic interaction of acidification and P limitation, the projected $N_2$ fixation by *Trichodesmium* was reduced more pronouncedly by 22.8 Tg N yr⁻¹ by the end of this century (Fig. 4c). The reduction was most marked in the western Pacific and the northern Indian Oceans where P is predicted to limit diazotroph activity[43]. It should be noted that these large-scale effects of OA and P limitation on *Trichodesmium* $N_2$ fixation illustrated are a first-order estimate, largely representing their relative strength in different oceanic regions, and uncertainties in the estimates due to the offline calculations and the assumption of a linear function of OA impacts (see the "Methods" section) are likely. Nevertheless, the evaluations based on our experimental findings suggest that $N_2$ fixation by *Trichodesmium* will probably decrease and its distribution will change in the future ocean due to interactions between P availability and acidification, although changes in other environmental factors (e.g., light, temperature, and Fe) in the future ocean may also affect the growth of *Trichodesmium* and modulate its response to ocean acidification[6,44,45].

Ocean acidification as a result of the dissolution of anthropogenic $CO_2$ in the ocean is anticipated to affect the globally important $N_2$ fixer *Trichodesmium*[6,8]. Our study demonstrates that the adverse effects of acidification on $N_2$ fixation by this prominent diazotroph can be

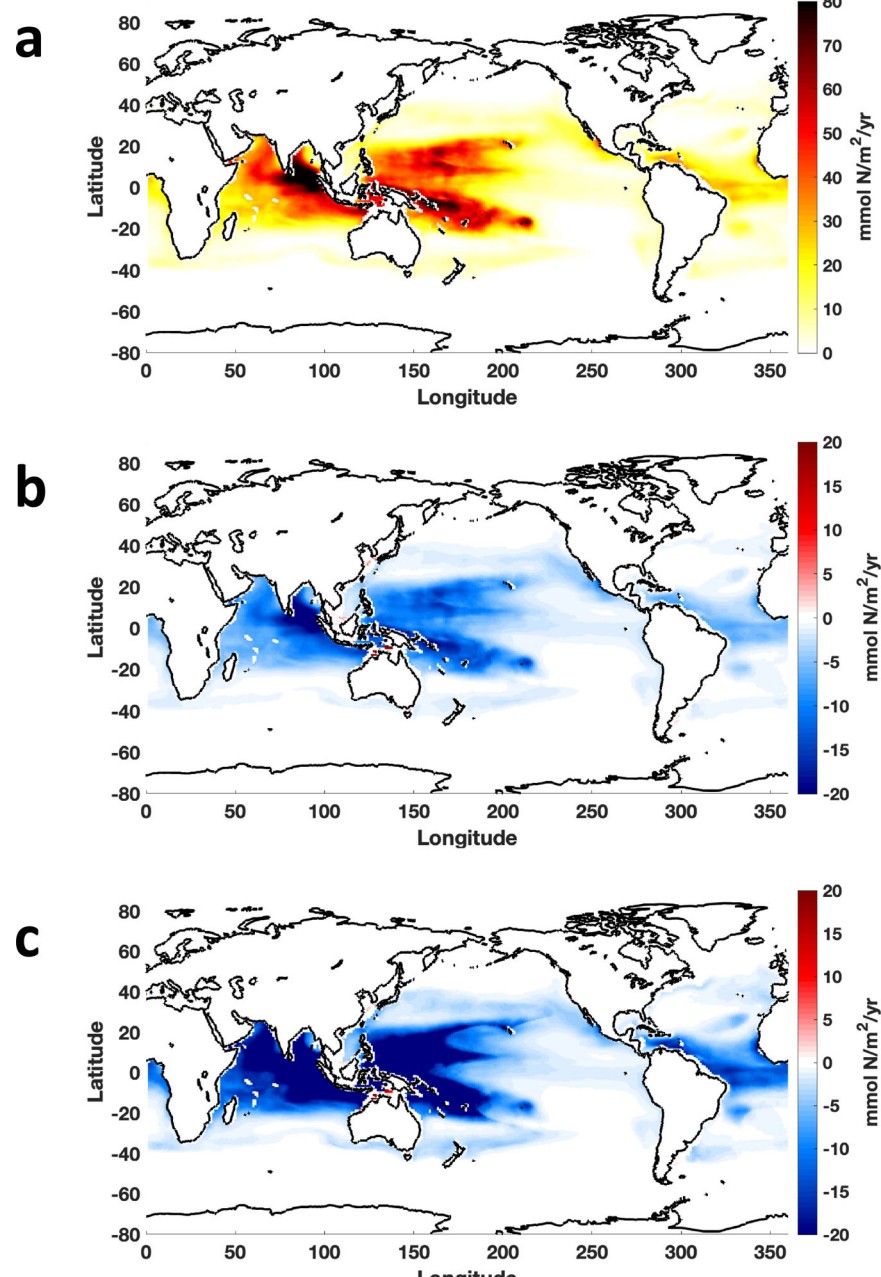

**Fig. 4 | Modeled changes in global marine N₂ fixation contributed by *Trichodesmium*.** **a** Predicted global *Trichodesmium* N₂ fixation (47.4 Tg N yr⁻¹) in 2081–2100 under the IPCC RCP 8.5 scenario using the Community Earth System Model with estimated *Trichodesmium* contribution (see the "Methods" section). **b**, **c** Additional changes in predicted *Trichodesmium* N₂ fixation due to ocean acidification alone (−11.3 Tg N yr⁻¹) (**b**) and to ocean acidification and P limitation combined (−22.8 Tg N yr⁻¹) (**c**) in 2081–2100. Data are visualized using MATLAB_R2018b.

magnified by P limitation that presumably will intensify along with seawater acidification in the future oligotrophic surface oceans[10]. It should be noted that although DOP was not the added P source in our study, it can represent a significant P source for *Trichodesmium* growth in $PO_4^{3-}$-limited surface waters[19,20]. Therefore, the effects of OA on DOP utilization and thus N₂ fixation by *Trichodesmium* certainly warrant future investigation. Nevertheless, the resulting decrease in cellular PON:POP ratio (48:1 and 31:1, respectively, under ambient and acidified conditions) observed here indicates that *Trichodesmium* may fix less N per unit of P in response to seawater acidification. Together, these results suggest that ocean acidification may lead to a decline in the supply of new nitrogen by *Trichodesmium* to the oceans, particularly in the vast P-limited regions[9]. Given the pivotal role *Trichodesmium* plays

in the marine nitrogen cycle, this perturbation is significant for ocean biology and biogeochemistry.

## Methods

### Laboratory experiments

**Culturing.** The marine cyanobacterium *Trichodesmium erythraeum* IMS101 was obtained from the National Center for Marine Algae and Microbiota (Maine, USA) and was grown in Aquil-tricho medium prepared with 0.22 μm-filtered and microwave-sterilized oligotrophic South China Sea surface water[6]. The medium was enriched with various concentrations of chelexed and filter-sterilized NaH₂PO₄ as where indicated, and filter-sterilized vitamins and trace metals buffered with 20 μM EDTA[6]. The cultures were unialgal, and although they were not

axenic, sterile trace metal clean techniques were applied for culturing and experimental manipulations. *T. erythraeum* was pre-adapted to low P condition by semi-continuously culturing at 0.5 μM $PO_4^{3-}$ and at two pCO₂ levels (400 and 750 μatm) for more than one year. To start the chemostat culture, three replicates per treatment were grown in 1-L Nalgene® magnetic culture vessels (Nalgene Nunc International, Rochester, NY, USA), in which the cultures were continuously mixed by bubbling with humidified and 0.22 μm-filtered CO₂–air mixtures and stirring using a suspended magnetic stir bar. The reservoirs contained Aquil-tricho medium with 1.2 μM $NaH_2PO_4$, which was delivered to the culture vessels using a peristaltic pump (Masterflex® L/S®, USA) at the dilution rate of 0.2 d⁻¹. In all experiments, cultures were grown at ;27 °C and -80 μmol photons m⁻² s⁻¹ (14 h:10 h light–dark cycle) in an AL-41L4 algae chamber (Percival). The concentration of Chlorophyll *a* (Chl*a*) was monitored daily in the middle of the photoperiod as an indicator of biomass. When the Chl*a* concentration remained constant for more than one generation, the system was considered to have reached steady-state, and was maintained for at least another four generations prior to sampling for further analysis.

**Carbonate chemistry manipulation.** pCO₂/pH of seawater media in the culture vessels and in the reservoir was controlled by continuously bubbling with humidified and 0.22 μm-filtered CO₂-air mixtures generated by CO₂ mixers (Ruihua Instrument & Equipment Ltd.). During the experimental period, the $pH_T$ (pH on the total scale) of media was monitored daily using a spectrophotometric method[46]. The dissolved inorganic carbon (DIC) of media was analyzed by acidification and subsequent quantification of released CO₂ with a CO₂ analyzer (LI 7000, Apollo SciTech). Calculations of alkalinity and pCO₂ were made using the CO2Sys program[47], based on measurements of $pH_T$ and DIC, and the carbonate chemistry of the experiments are shown in Supplementary Table 1.

**Chla concentration and cell density and size.** Chl*a* concentration was measured daily following Hong et al.[6]. Briefly, *T. erythraeum* was filtered onto 3 μm polycarbonate membrane filters (Millipore), followed by heating at 65 °C for 6 min in 90% (vol/vol) methanol. After extraction the filter was removed and cell debris were spun down via centrifugation (5 min at 20,000×*g*) before spectrophotometric analysis. Cell density and the average cell length and width were determined at regular intervals when the chemostat cultures reached steady-state using ImageJ software. Photographs of *Trichodesmium* were taken using a camera (Canon DS126281, Japan) connected with an inverted microscope (Olympus CKX41, Japan). Total number and length of filaments in 1 mL of culture were measured, and the cell number of -20 filaments was counted. The average length of cells was obtained by dividing the total length of the 20 filaments by their total cell number. The cell density of the culture was then calculated by dividing the total length of filaments in 1 mL culture by the average cell length. The average cell width was determined by measuring the width of around 1000 cells in each treatment.

**Elemental composition.** To determine particulate organic C (POC) and N (PON), at the end of the chemostat culturing *T. erythraeum* cells were collected on pre-combusted 25 mm GF/F filters (Whatman) and stored at −80 °C. Prior to analysis, the filters were dried overnight at 60 °C, treated with fuming HCl for 6 h to remove all inorganic carbon, and dried overnight again at 60 °C. After being packed in tin cups, the samples were subsequently analyzed on a PerkinElmer Series II CHNS/ O Analyzer 2400.

Particulate organic P (POP) was measured following Solorzano et al.[48]. Cells were filtered on pre-combusted 25 mm GF/F filters and rinsed twice with 2 mL of 0.17 M Na₂SO₄. The filters were then placed in combusted glass bottles with the addition of 2 mL of 0.017 M MgSO₄, and subsequently evaporated to dryness at 95 °C and baked at 450 °C

for 2 h. After cooling, 5 mL of 0.2 M HCl was added to each bottle. The bottle was then tightly capped and heated at 80 °C for 30 min, after which 5 mL Milli-Q H₂O was added. Dissolved phosphate from the digested POP sample was measured colorimetrically following the standard phosphomolybdenum blue method.

**C uptake and N₂ fixation rates.** Rates of short-term C uptake were determined at the end of the chemostat culturing. 100 μM NaH¹⁴CO₃ (PerkinElmer) was added to 50 mL of cultures in the middle of the photoperiod, which was then incubated for 20 min under the growth conditions. After incubation, the samples were collected onto 3 μm polycarbonate membrane filters (Millipore), which were then washed with 0.22 μm-filtered oligotrophic seawater and placed on the bottom of scintillation vials. The filters were acidified to remove inorganic C by adding 500 μL of 2% HCl. The radioactivity was determined using a Tri-Carb 2800TR Liquid Scintillation Analyzer (PerkinElmer). Rates of N₂ fixation (nitrogenase activity) were measured in the middle of the photoperiod for 2 h by the acetylene reduction assay[49], using a ratio of 4:1 to convert ethylene production to N₂ fixation.

**Soluble reactive phosphate (SRP) analysis.** When the chemostat cultures reached a steady-state, SRP concentrations in the culture vessels were measured at regular intervals, using the classic phosphomolybdenum blue (PMB) method with an additional step to enrich PMB on an Oasis HLB cartridge[50]. Briefly, 100 mL of GF/F filtered medium sample was fortified with 2 mL of ascorbic acid (100 g L⁻¹) and 2 mL of mixed reagent (MR, the mixture of 100 mL of 130 g L⁻¹ ammonium molybdate tetrahydrate, 100 mL of 3.5 g L⁻¹ potassium antimony tartrate, and 300 mL of 1:1 diluted H₂SO₄), and then mixed completely. After standing at room temperature for 5 min, the solution was loaded onto a preconditioned Oasis HLB cartridge (3 cm³/60 mg, P/N: WAT094226, Waters Corp.) via a peristaltic pump, and then 1 mL eluent solution (0.2 M NaOH) was added to elute the sample into a cuvette, to which 0.06 mL of MR and 0.03 mL of ascorbic acid solution was added to fully develop PMB. Finally, the absorbance of PMB was measured at 700 nm using a spectrophotometer.

**Alkaline phosphatase (AP) activity.** AP activities were measured in the middle of the photoperiod using *p*-nitrophenylphosphate (pNPP) as a substrate[51]. Briefly, 5 mL of culture was incubated with 250 μL of 10 mM pNPP, 675 μL of Tris-glycine buffer (50 mM, pH 8.5) and 67.5 μL of 1 mM MgCl₂ for 2 h under growth conditions. The absorbance of formed *p*-nitrophenol (pNP) was measured at 410 nm using a spectrophotometer.

**PolyP analysis.** At the end of the chemostat culturing, *T. erythraeum* cells were filtered in the middle of the photoperiod onto 3 μm polycarbonate membrane filters (Millipore), flash frozen in liquid nitrogen, and stored at −80 °C until analysis. PolyP was quantified fluorometrically following Martin and Van Mooy[22] and Martin et al.[23]. Briefly, samples were re-suspended in 1 mL Tris buffer (pH 7.0), sonicated for 30 s, immersed in boiling water for 5 min, sonicated for another 30 s, and then digested by 10 U DNase (Takara), RNase (2.5 U RNase A + 100 U RNase T1) (Invitrogen) and 20 μl of 20 mg mL⁻¹ proteinase K at 37 °C for 30 min. After centrifugation for 5 min at 14,000×*g*, the supernatant was diluted with Tris buffer according to the range of standards curve, stained with 60 μL of 100 μM 4, 6-diamidino-2-phenylindole (DAPI) per 500 μL of samples, incubated for 7 min and then vortexed. The samples were then loaded onto a black 96-well plate and the absorption of fluorescence at an excitation wavelength of 415 nm and emission wavelength of 550 nm was measured using a PerkinElmer EnSpire® Multimode Plate Reader. PolyP standard (sodium phosphate glass Type 45) was purchased from Sigma-Aldrich. This method gives a relative measure of polyP concentration[23] that is expressed as femto-equivalents of the standard per cell (feq cell⁻¹).

**Cellular ATP measurement.** Cellular ATP contents were determined when the chemostat cultures reached a steady state. *T. erythraeum* cells were collected in the middle of the photoperiod using an ATP Assay Kit (Beyotime Biotechnology, Shanghai, China) according to the manufacturer's instructions. Briefly, the sample was lysed and centrifuged, and the supernatant (100 μL) was mixed with ATP detection working reagent (100 μL) and loaded onto a black 96-well plate. The luminescence was measured using a PerkinElmer EnSpire® Multimode Plate Reader.

**Intracellular metabolites measurements.** NAD(H), NADP(H), and Glu were measured at the end of the chemostat culturing, using the liquid chromatography-tandem quadrupole mass spectrometry (LC–MS/MS) method modified from Luo et al.[52]. Briefly, *T. erythraeum* cells were gently filtered at the middle of photoperiod onto 3 μm polycarbonate membrane filters (Millipore), rapidly suspended in −80 °C precooled methanol-water (60%, v/v) mixture. After being kept in −80 °C freezer for 30 min, the sample was sonicated for 30 s, centrifuged at 12,000×$g$ and 4 °C for 5 min, and the supernatant was filtered through a 0.2 μm filter (Jinteng®, China) and stored at −80 °C for further LC–MS/MS analysis.

A 2.0 × 50 mm Phenomenex® Gemini 5u C18 110 Å column (particle size 5.2 μm, Phenomenex, USA) was used for the analysis. The mobile phases consisted of two solvents: mobile phase A (10 mM tributylamine aqueous solution, pH 4.95 with 15 mM acetic acid) and mobile phase B (100% methanol), which were delivered using an Agilent 1290 UPLC binary pump (Agilent Technologies, Palo Alto, CA, USA) at a flow rate of 200 μL min$^{-1}$, with a linear gradient program implemented as follows: hold isocratic at 0% B (0–2 min); linear gradient from 0% to 85% B (2–28 min); hold isocratic at 0% B (28–34 min). The effluent from the LC column was delivered to an Agilent 6490 triple-quadrupole mass spectrometer, equipped with an electrospray ionization source operating in negative-ion mode. NAD, NADH, NADP, NADPH, and Glu were monitored in the multiple reaction monitoring modes with the transition events at $m/z$ 662.3 > 540, 664.3 > 79, 742 > 620, 744 > 79, and 147 > 84, respectively.

### RNA extraction, library preparation, and sequencing

At the end of the chemostat culturing, *T. erythraeum* was collected in the middle of the photoperiod by filtering onto 3 μm polycarbonate membrane filters (Millipore), flash frozen in liquid nitrogen and stored at −80 °C until extraction. Total RNA was extracted using TRIzol® Reagent (Invitrogen) combined with a physical cell disruption approach by glass beads according to the manufacturer's instructions. Genomic DNA was removed thoroughly by treating it with RNAase-free DNase I (Takara, Japan). Ribosomal RNA was removed from a total amount of 3 μg RNA using Ribo-Zero rRNA Removal kit (Illumina, USA). Subsequently, cDNA libraries were generated according to the manufacturer's protocol of NEBNext® Ultra™ Directional RNA Library Prep Kit for Illumina® (NEB, USA). The quality of the library was assessed on the Agilent Bioanalyzer 2100 system (Agilent Technologies, CA, USA). Libraries were sequenced on an Illumina Hiseq 2500 platform, yielding 136-bp paired-end reads.

**RNA-Seq bioinformatics.** Clean reads were obtained from raw data by removing reads containing adapter, ploy-N and low-quality read. Qualified sequences were mapped to the *Trichodesmium erythraeum* IMS101 genome (https://www.ncbi.nlm.nih.gov/nuccore/NC_008312.1) by using Bowtie2-2.2.3[53]. Differential expression analysis for high/low pCO₂ with P limitation was performed using the DESeq2 R package[54]. The resulting $p$-values were adjusted using Benjamini and Hochberg's approach for controlling the false discovery rate. Genes with an adjusted $p$-value < 0.05 were assigned as significantly differentially expressed. The data are deposited in NCBI's Gene Expression Omnibus and are accessible through GEO Series accession number GSE181428.

GO and KEGG enrichment analyses of differentially expressed genes were implemented by the GOseq R package and KOBAS software respectively. GO terms with corrected $p$-value < 0.05 were considered significantly enriched by differentially expressed genes.

**Quantitative polymerase chain reaction (qPCR) analysis and standard preparation.** Extracted RNA was treated with DNase and then reverse-transcribed using random primers by M-MLV reverse transcriptase (BGI, China) to generate cDNA. The primer sequences of each gene were obtained from the literature or designed at the Genscript website and checked for validity using the Primer-Blast tool in NCBI (Supplementary Table 7). The standards for qPCR were generated as described previously[6]. Briefly, cDNA amplicon of the interested genes was PCR amplified, separated by gel electrophoresis, purified, inserted into pMD 18-T vectors (Takara), and used to transform DH5a *Escherichia coli* competent cells. Sequenced plasmid DNA from positive clones was purified and then quantified using the Qubit DNA HS Assay kit (Invitrogen).

All qPCR reactions were carried out on a fluorescent quantitative instrument CFX 96 TOUCH (Bio-Rad Laboratories). An SYBR Green I master mix (Zhishan Biotech) was used for qPCR in 20 μL reactions containing ~5 μL of diluted cDNA template, 0.4 mM dNTPs, 200 nM of each primer, and 0.05 U Taq polymerase (Tiangen Biotech). The following PCR reactions program was applied: 95 °C for 3 min, followed by 39 cycles of 95 °C for 15 s, 60 °C for 30 s, and 72 °C for 30 s. Standards corresponding to between 10³ and 10⁸ copies per well were amplified on the same 96-well plate as the cDNA generated from experimental materials. The amplification efficiencies of PCR were always between 90% and 103% with $R^2$ values >0.99. To correct for differences in cDNA synthesis efficiency, the abundance of each transcript was normalized to the abundance of the housekeeping gene *FtsZ* (cell division protein) transcript.

**Calculating P contents in RNA and genomic DNA.** Extracted total RNA was exactly quantified using the Qubit RNA HS Assay kit (Invitrogen). According to the approximate average M.W. of nucleotide within polynucleotide, i.e., 320.5 g mol$^{-1}$ (Thermo Fisher Scientific), the P contents in RNA were estimated. According to the genome size of *T. erythraeum* IMS101 (i.e., 7,750,108 bp) and taking into account polyploidy in *Trichodesmium* (i.e., assuming 100 genome copies per cell under P-limited conditions)[55], the P contents in genomic DNA was estimated.

**Statistical analysis.** The statistical significance of differences between ambient and acidified treatments was analyzed by two-tailed paired Student's $t$-test using SigmaPlot 12.5 (Systat Software, Inc). A significance level of $p < 0.05$ was applied.

### Field experiments

A total of four phosphate amendment experiments and seven ocean acidification experiments were conducted aboard the R/V *Dongfanghong* 2 and R/V *Tan Kah K*ee during several cruises to the northern South China Sea between May 2016 and August 2018 (Supplementary Fig. 4).

### Phosphate amendment experiments

**Experimental setup.** The incubation experiments were carried out at stations TM-4, TM-5, S1, and SK2 (Supplementary Fig. 4). Trace metal-clean seawater was collected using a towed fish system in which surface seawater (~5 m) was pumped through Teflon tubing into 10-L polycarbonate carboys directly, using a Teflon diaphragm pump (Sandpiper, USA). Trace metal-clean technique was used in setting up and sampling the experiments. All materials coming in contact with the incubation water were acid-washed in a class-100 cleanroom before use. For the phosphate-amended treatments, NaH₂PO₄ (chelexed and

filter-sterilized) was added at a final concentration of 100 nM. For all experiments, triplicate carboys were incubated for 3 days in on-deck, flow-through incubators screened with neutral density screening to ~45% of sea surface irradiance, and subsampling was done in a laminar flow hood.

**Sample collection for Trichodesmium genes sequencing and sphX gene transcription.** After incubation, 3–4 L of seawater from each one of the triplicate carboys was collected by filtration under low vacuum pressure onto 47 mm 0.2 μm polycarbonate membrane filters (Millipore) for subsequent DNA and RNA extraction. Nucleic acid filtrations were typically completed within 45 min. All filters were flash-frozen in liquid nitrogen and kept at −80 °C until extraction.

### Ocean acidification experiments

**Experimental setup.** The incubation experiments were carried out using either 2–4 L polycarbonate bottles or 10–20 L polycarbonate carboys (Nalgene Labware) at 7 stations including TM-4 and TM-5 where the phosphate amendment experiments were conducted (Supplementary Fig. 4). At station OA-5, a *Trichodesmium* bloom occurred. For the experiments at stations, OA-2–OA-5, non-trace metal-clean near-surface seawater (~5 m) was collected using either a Teflon diaphragm pump or a Sea-bird CTD-General Oceanic rosette sampler with GO-Flo bottles. For the rest of the experiments (i.e., at stations TM-3–TM-5), the trace metal-clean technique was used in seawater sampling and experimental setup, as described above. For all the experiments, triplicate carboys or bottles were incubated for 1–3 days in on-deck, flow-through incubators screened to ~45% of sea surface irradiance.

**Carbonate chemistry manipulation.** For the experiments at station OA-5, ultra-pure HCl and NaOH were used to adjust seawater carbonate chemistry. For the experiments at other stations, seawater carbonate chemistry was manipulated by gently bubbling with 0.22 μm-filtered air or $CO_2$–air mixture generated by $CO_2$ mixers (Ruihua Instrument & Equipment Ltd.). pH was measured using a pH electrode (Eutech pH 110 meter with Eutech ECFC7352901B probe) calibrated with National Institute of Standards and Technology pH standard buffers. Intercalibrations between the electrode measurements and spectrophotometric pH measurements[46] were made on seawater samples to arrive at $pH_T$ (pH on the total scale). The measurements of dissolved inorganic carbon (DIC) concentration and the calculations of alkalinity and $pCO_2$ were done as described above. The carbonate chemistry of the different experiments is shown in Supplementary Table 6.

**$N_2$ fixation rates.** $N_2$ fixation rates were measured using the $^{15}N_2$ gas dissolution method[56]. Briefly, 0.22 μm-filtered surface seawater was degassed as described in Shiozaki et al.[57]. After that, 10 mL 98% pure $^{15}N_2$ gas (Cambridge Isotope Laboratories) was injected into a gas-tight plastic bag containing 1 L of the degassed seawater and allowed to fully equilibrate before use. The percentage of $^{15}N_2$ in the $^{15}N_2$-enriched seawater was validated using a GasBench-IRMS[58]. Following 1–3 days of incubation, a sub-sample from each incubation carboy was transferred into a polycarbonate bottle, and then $^{15}N_2$-enriched seawater was added with the enriched water constituting approximately 2.6% of the total sample volume. Bottles were then returned to the flow-through incubators under the same irradiance of each experiment and incubated for 24 h. After the incubation, particulate matter in seawater from each bottle was filtered onto 25 mm pre-combusted GF/F filters (Millipore). Particulate organic matters collected from the same incubation carboy but without $^{15}N_2$ enrichment were filtered to determine $^{15}N$-PON natural abundance. All filter samples were stored at −20 °C immediately after collection. In the shore-based laboratory, sample filters were dried and analyzed using a Flash HT 2000 elemental

analyzer coupled with a Thermo Finnigan Delta V Plus isotope ratio mass spectrometer. The rates of $N_2$ fixation were calculated according to Mohr et al.[56].

**Soluble reactive phosphate (SRP) analysis.** The SRP concentrations were measured using the method as described above in I. Laboratory experiments.

**Sample collections for nifH abundance and Trichodesmium ppnK gene transcription.** To analyze *nifH* gene abundance, at most stations, 3–4 L (500 mL at station OA-5) of seawater from each one of the triplicate carboys was collected following the incubation by filtration under low vacuum pressure onto 47 mm 0.2 μm polycarbonate membrane filters (Millipore). At station OA-2, 4.5 L of the surface seawater was directly collected for *nifH* gene analysis without incubation. For *Trichodesmium ppnK* gene transcription analysis, 3–4 L of seawater was subsampled from the incubations at stations TM-3, TM-4, TM-5, OA-3, OA-4, and OA-5. All filters were flash-frozen in liquid nitrogen and kept at −80 °C until DNA/RNA extraction.

**DNA extraction.** To extract DNA, membrane filters were cut into pieces under sterile conditions and then placed in tubes containing 800 μL of sucrose lysis buffer (40 mM EDTA, 50 mM Tris–HCl, and 0.75 M sucrose) for beads beating with 0.1 mm and 0.5 mm glass beads. The cells were broken, agitated for 3 min inside a Fast Prep machine (MP Biomedicals, USA), and frozen in liquid nitrogen 3 times. 5 μL of lysozyme (100 mg mL$^{-1}$) was then added and the sample was incubated for 1 h at 37 °C. After incubation, the lysates were transferred into a 2-mL Eppendorf tube. Proteins were digested by incubating with 1% sodium dodecyl sulfate (SDS) and proteinase K (250 μg mL$^{-1}$) at 55 °C for 2 h, and were removed by centrifugation at 12,000×$g$ for 20 min at 4 °C after being treated with an equal volume of phenol:chloroform:isoamyl alcohol (25:24:1) containing 5 M NaCl. As a result, the sample was separated into three layers. The top aqueous layer that contained genomic DNA was transferred into a new tube, added an equal volume of chloroform: isoamyl alcohol (24:1), and centrifuged at 12,000×$g$ for 20 min at 4 °C. Genomic DNA was purified by precipitation with 100% isopropanol at −20 °C overnight, followed by washing with 70% ethanol and air-drying. The genomic DNA was then eluted into 50 μL TE buffer and stored at −20 °C.

**RNA extraction and reverse transcription-PCR.** Total RNA was extracted using the RNeasy Mini Kit (Qiagen, Hildern, Germany) according to manufacturer instructions, with a minor modification to the cell disruption step. Briefly, RLT buffer with 1% of $β$-Mercaptoethanol and RNA-clean glass beads (0.1 mm diameter) were added and samples were vortexed with a Fast Prep machine (MP Biomedicals, USA). The resulting lysate was processed following the manufacturer's instructions, including on-column DNase digestion (RNase-free DNase Kit, Qiagen), and then RNA was eluted in RNase-free water. The concentration of RNA was determined using the Qubit RNA HS Assay kit (Invitrogen). Reverse transcription was carried out using M-MLV reverse transcriptase in a 20 μL reaction volume containing 150 ng of random primers, 1 mM dNTP mix, and 10 mM DTT. Complimentary DNA (cDNA) was stored at −20 or −80 °C.

**Cloning and sequencing.** According to genome sequences of *T. erythraeum* IMS101 and *T. thiebautii* H9-4, the degenerate primers used to amplify the high-affinity phosphate binding protein gene *sphX* and the inorganic polyphosphate/ATP_NAD kinase gene *ppnk* from *Trichodesmium* species were designed (Supplementary Table 7). The *sphX* (783 bp) and *ppnK* (620 bp) genes were amplified from samples collected at stations TM-4 and TM-5. The amplified products were separated by gel electrophoresis, purified, and cloned for sequencing. Forty positive clones were sequenced for each gene from each station.

The cloned sequences were aligned with the target gene sequence in *T. erythraeum* IMS101 and *T. thiebautii* H9-4 genomes using Mega software version 7.0 and were then used to generate a maximum-likelihood phylogenetic tree (Supplementary Fig. 5). The *sphX* and *ppnk* sequences produced in this study have been submitted to GenBank and are accessible through GenBank accession numbers MZ749754 – MZ749900.

**Quantitative qPCR analysis.** The clade-specific primers of house-keeping gene *rnpB* and the Tery-specific *sphX* primers were listed in Supplementary Table 7. According to our alignments, the clade-specific *ppnK* primers and the Tten-specific *sphX* primers were designed at the Genscript website and checked for validity using the Primer-Blast tool in NCBI (Supplementary Table 7). Standard curves for qPCR were constructed using cloned target DNA fragments. Briefly, cDNA amplicon of the genes of interest was PCR amplified, separated by gel electrophoresis, purified, and cloned. Sequenced plasmid DNA from positive clones was purified and quantified using the Qubit DNA HS Assay kit (Invitrogen). All qPCR reactions were carried out on a fluorescent quantitative instrument CFX 96 TOUCH (Bio-Rad, Singapore). A SYBR Green I master mix (Zhishan Biotech) was used for qPCR in 20 μL reactions containing ~5 μL of diluted cDNA template, 0.4 mM dNTPs, 200 nM of each primer, and 0.05 U Taq polymerase (Tiangen Biotech). The following PCR reactions program was applied: 95 °C for 3 min, followed by 39 cycles of 95 °C for 15 s, 60 °C for 30 s and 72 °C for 30 s. Using 10-fold increments, the standards corresponding to between $10^0$ and $10^9$ copies per well were amplified on the same 96-well plate as the cDNA generated from experimental materials. The amplification efficiencies of PCR were always between 90 and 95% with $R^2$ values >0.99. The specificity of the qPCR reactions was confirmed by melting curve analysis, agarose gel electrophoresis, and sequencing analysis. Relative expression of target genes versus *rnpB* was calculated by dividing the copy numbers of the target gene by the copy numbers of *rnpB* quantified in the same sample.

**nifH Abundance.** The qPCR analysis was targeted on the *nifH* phylotypes of *Trichodesmium* spp., *Richelia* sp. (Het-1), unicellular cyanobacteria UCYN-A1, UCYN-A2 and UCYN-B, and a gamma-proteobacterium (γ-24774A11) using previously designed primers[59–61]. Probes were 5′-labeled with the fluorescent reporter FAM (6-carboxyfluorescein) and 3′-labeled with TAMRA (6-carboxytetramethylrhodamine) as a quenching dye. The *nifH* standards were obtained by cloning the environmental sequences from previous samples of the South China Sea. The DNA concentrations of *nifH* standards were determined by Quant-iTTM Picogreens® dsDNA Reagent and Kits (Invitrogen) using a Fluoroskan Ascent FL fluorescence microplate reader (Thermo Scientific). Quantitative PCR analysis was carried out as described previously[60] with slight modifications. qPCR reactions were run in triplicate for each environmental DNA sample and each standard, using the following thermal cycle program: 50 °C for 2 min, 94 for 10 min, followed by 49 cycles of 95 °C for 15 s, 60 °C for 1 min. The 20 μL reactions contained environmental cDNA or standard (1 μL), forward and reverse primers, and probes. Standards corresponding to between $10^1$ and $10^9$ copies per well were amplified on the same 96-well plate. The copy numbers of the target genes in the environmental samples were calculated from the standard curve.

**Statistical analysis.** Statistical significance of differences between ambient and acidified treatments was tested both on data from each individual experiment and on data from all experiments by one-tailed paired Student's *t*-test using SigmaPlot 12.5 (Systat Software, Inc.). A significance level of $p < 0.05$ was applied.

## Model estimations

**Contribution to global oceanic N₂ fixation by *Trichodesmium*.** N₂ fixation rate by each diazotrophic group was estimated by the product of *nifH*-based diazotrophs' cell abundance and their cell-specific N₂ fixation rate:

$$N_2 \text{ fixation rate of individual diazotrophic group (fmol NL}^{-1}\text{ h}^{-1}) = nifH \text{ abundance}/ \text{conversion factor of } nifH \text{ copies to cell count} \times \text{cell specific } N_2 \text{ fixation rate} \quad (1)$$

where *nifH* abundance = *nifH* gene copies L$^{-1}$, conversion factor of *nifH* copies to cell count = *nifH* gene copies cell$^{-1}$, and cell specific N₂ fixation rate = fmol N cell$^{-1}$ h$^{-1}$.

We first used a machine learning algorithm (build in MATLAB_R2021a)−random forest−to estimate the *nifH* gene abundance of *Trichodesmium*, UCYN-A, UCYN-B (*Crocosphaera*), and *Richelia* (diatom-diazotroph associations) in the global ocean based on Tang and Cassar[42]. Briefly, field-observed volumetric *nifH* gene abundance was matched to contemporaneous, climatological, or modeled environmental predictors including temperature, salinity, dissolved oxygen, nitrate, phosphate, photosynthetically available radiation (PAR), and iron. Climatological temperature, salinity, dissolved oxygen, nitrate, and phosphate data were obtained from World Ocean Atlas 2018[62]. PAR was measured by SeaWiFS and MODIS satellites, and dissolved iron concentration in the global ocean was modeled by CESM2[63]. After training random forest models with these environmental properties (Supplementary Fig. 6), we estimated the volumetric *nifH* gene abundance of four diazotrophic groups in the global ocean using the environmental factors. The conversion factor of *nifH* copies to cell counts and the cell-specific N₂ fixation rate were obtained from the existing literature for each diazotrophic group (Supplementary Table 8). Average values for each parameter were used in the rate estimate. N₂ fixation was assumed to be active over 12 h daily. Overall, *Trichodesmium* accounts for approximately 35% of the global oceanic N₂ fixation (Supplementary Fig. 7). We acknowledge the large uncertainties associated with our estimates due to the variation in parameters used in the calculation (e.g., variation in the conversion factor from *nifH* gene abundance to cell count[55,64] and cell-specific N₂ fixation rate[65]). Nevertheless, our estimated total contribution of *Trichodesmium* to the global oceanic N₂ fixation, in particular its dominance in tropical and subtropical oceans, is generally within the range of the previous estimates[2,66,67]. More targeted observations on *Trichodesmium* will be required to better constrain its contribution to the global N₂ fixation.

**Effects of ocean acidification and P limitation on N₂ fixation by Trichodesmium.** We estimated changes in the global oceanic N₂ fixation under the impacts of ocean acidification and/or P limitation by the end of 21st century under the representative concentration pathway (RCP) 8.5 emission scenario. The global oceanic N₂ fixation was simulated using the biogeochemical elemental cycling (BEC) Model embedded in the Community Earth System Model version 1 (CESMv1)[68], which has been described in detail previously[69,70]. Further calculations were conducted using code written with MATLAB_R2018b. As in most Earth system models, there is only one diazotrophic functional group representing all N₂ fixers in BEC. Therefore, N₂ fixation by *Trichodesmium* is calculated by multiplying the BEC-simulated N₂ fixation by the estimated contribution of *Trichodesmium* (Supplementary Fig. 7). Note that impacts of acidification on N₂ fixation were not represented in the BEC. An offline mode of BEC was then combined with our laboratory results to illustrate the potential magnitude of impacts of ocean acidification and/or P limitation on the global N₂ fixation by *Trichodesmium*.

First, we estimated the impacts of ocean acidification without considering P-limitation. As shown in a previous study[8], when P is not limiting ocean acidification only affects N₂ fixation, not C biomass. Assuming N₂ fixation rates declined linearly with [H$^+$] in acidified

waters, we interpolated and/or extrapolated impacts of ocean acidification based on the laboratory results and simulated pH for each model grid of CESMv1 from 2081 to 2100:

$$NF_{Tr}^{oa} = NF \times R_{Tr} \times \left(100\% + oa \times \frac{[H^+]_s - [H^+]_0}{[H^+]_1 - [H^+]_0}\right) \quad (2)$$

where $NF_{Tr}^{oa}$ is $N_2$ fixation by *Trichodesmium* at simulated $[H^+]$ ($[H^+]_s$) in the grid under the RCP 8.5 scenario, NF is the BEC-simulated $N_2$ fixation, $R_{Tr}$ is the contribution of *Trichodesmium* to $N_2$ fixation (Supplementary Fig. 7), and oa of −18.4% is from a previous study[8] in which C biomass-specific $N_2$ fixation rate of *Trichodesmium* declined by 18.4% as seawater $[H^+]$ increased from $[H^+]_0$ ($10^{-8.01}$) to $[H^+]_1$ ($10^{-7.81}$) when P is not limiting (Supplementary Table 2). Meanwhile, the simulated climatology of *Trichodesmium* $N_2$ fixation without ocean acidification impacts (i.e., $NF_{Tr} = NF \times R_{Tr}$) from 2081 to 2100 is used as the reference. Comparison of the integrated $NF_{Tr}^{oa}$ with the reference value, i.e., $\sum NF_{Tr}^{oa} - \sum NF_{Tr}$, shows impacts of ocean acidification on the global $N_2$ fixation by *Trichodesmium*.

Next, we estimated the impacts of ocean acidification on *Trichodesmium* with the synergistic impacts of P limitation being considered. Our laboratory experiments showed that ocean acidification affects both C biomass and $N_2$ fixation under P limitation, which was taken into account in our estimations. The limitation strength of various nutrients was calculated in the BEC model, which can be used to identify waters in which *Trichodesmium* growth was limited by P availability. Since *Trichodesmium* can use both DIP and DOP for their growth in the model, both forms of P are considered in the limiting factor calculations based on Michaelis−Menten kinetics:

$$V_i^{PO_4} = \frac{PO_4/K_i^{PO_4}}{1 + PO_4/K_i^{PO_4} + DOP/K_i^{DOP}};$$
$$V_i^{DOP} = \frac{DOP/K_i^{DOP}}{1 + PO_4/K_i^{PO_4} + DOP/K_i^{DOP}}; \text{ and} \quad (3)$$
$$V_i^{P} = V_i^{PO_4} + V_i^{DOP}.$$

where $V_i^{P}$ is the limiting factor for P, and $K_i^{PO_4}$ is about 5 times lower than $K_i^{DOP}$, representing that phosphate is a more preferred P source than DOP. The additional impacts of P limitation were then only applied to *Trichodesmium* in these P-limiting regions, while the impacts of acidification in P-replete waters were calculated similarly as in Eq. (2). Under P limitation, we assumed C biomass and $N_2$ fixation rate decreased linearly with $[H^+]$ based on our laboratory data:

$$NF_{Tr}^{oap} = NF \times R_{Tr} \times \left(100\% + oap_C \times \frac{[H^+]_s - [H^+]_0}{[H^+]_1 - [H^+]_0}\right)$$
$$\times \left(100\% + oap_N \times \frac{[H^+]_s - [H^+]_0}{[H^+]_1 - [H^+]_0}\right) \quad (4)$$

where $oap_C$ and $oap_N$ were set to −39.2% and −10.1%, respectively, based on our laboratory results in which C biomass (i.e., POC:POP) and C biomass-specific $N_2$ fixation rate of P-limited *Trichodesmium* decreased by 39.2% and 10.1%, respectively, under acidified conditions (Supplementary Table 2). The synergistic interaction of acidification and P limitation on the global $N_2$ fixation was calculated by $\sum NF_{Tr}^{oap} - \sum NF_{Tr}$.

The large-scale effects of ocean acidification and P limitation on *Trichodesmium* $N_2$ fixation illustrated here are first-order estimates largely representing their relative strength in different oceanic regions. Uncertainties existed in our estimates due to the assumption of a linear function and an offline calculation. However, simulations with explicit representations of various diazotrophic groups and their responses to acidification in Earth system models are not feasible

without further laboratory results to provide key information for model developments.

### Reporting summary

Further information on research design is available in the Nature Portfolio Reporting Summary linked to this article.

## Data availability

Source data are provided with this paper. The RNA-seq and DNA sequencing datasets generated in this study have been deposited in NCBI's Gene Expression Omnibus (GSE181428) and GenBank (MZ749754−MZ749900), respectively. The reference genome of *T. erythraeum* IMS101 is available in NCBI under accession code NC_008312.1. Contemporaneous, climatological, or modeled environmental data are available at World Ocean Atlas 2018 (https://accession.nodc.noaa.gov/NCEI-WOA18). Source data are provided with this paper.

## Code availability

The computer code of the numerical model used in this study can be found in Supplementary Methods.

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

## Acknowledgements

The authors thank B. Hopkinson for helpful discussions and Ruotong Jiang for assistance with graph plotting. The authors gratefully acknowledge the captains and crew of the R/V *Dongfanghong* 2 and R/V *Tan Kah Kee* for their help during the cruises. This work was supported by the National Science Foundation of China (41925026, 42076149, 41890802, 31861143022, and 41721005), and the XPLORER Prize from the Tencent Foundation to D.S. W.T. was supported by the Harry H. Hess Postdoctoral Fellowship from Princeton University.

## Author contributions

D.S. and H.H. designed the research. F.Z. and Z.W. performed the experiments. S.W., W.T., and Y.-W.L. performed numerical modeling. D.S., H.H., F.Z., Z.W., and S.A.K. analyzed data and wrote the manuscript. All authors discussed the results and commented on and revised the manuscript.

## Competing interests

The authors declare no competing interests.
