## [Peer Review File · Nature Communications]

Phosphate limitation intensifies negative effects of ocean acidification on globally important nitrogen fixing cyanobacteriumREVIEWER COMMENTS

Reviewer #1 (Remarks to the Author):

The authors have investigated the role of OA on cellular phosphorus dynamics under steady state conditions where the inorganic P supply is low. I commend the authors for their significant effort to achieve culture conditions with a steady state supply of inorganic phosphorus, compared to the more trodden path whereby cells are grown under semi-continuous or starvation conditions. To me, the authors' data suggests an enhanced cellular P quota much greater than they have reported (described below), and the prospect of poly-P accumulation is very interesting and deserves further exploration. I have been impressed with the authors' scholarship over the last 15 years, but I have reservations on some of the interpretations presented here. I hope the authors will find these comments useful and constructive towards an improved manuscript.

The authors report decreases in rates of C and N fixation due to OA conditions, and further report that the cellular PON and POC contents had also declined under OA compared to ambient conditions, "in accordance with the decreased fixation rates". I interpret their data differently, considering the biochemical relevance of the 'specific' rates. Comparing the appropriately normalized rates of C and N fixation to the cellular C and cellular N, respectively, suggests C and N fixation ~7% higher and ~14% lower under OA conditions (see attached table). I cannot help but wonder whether this 14% decrease is statistically significant. Similarly I estimate the C-normalized chlorophyll content of the smaller cells grown under OA conditions to be only 10% lower than that of ambient cells. In these regards, the emphasis and even the title of the manuscript may be reconsidered. If there is a silver lining here, it is that the POP:POC (ie cellular P quota) is actually 65% greater under OA conditions compared to ambient conditions (Note that Supplemental Table 1 is not consistent with data in Figs 1A and 1B; the reported acidified POC:POP appears to have been incorrectly calculated as the value from Figs 1A and 1B suggest a value of about 200:1 rather than 293:1 (mol:mol)).

If I understand the chemostat data properly (Supplemental Figure 1) the cells in both treatments were growing at the same rate. That the dilution rates of the chemostats under both conditions were set to 0.3 per day (0.3 litres per litre per day) and the cells were not washed out suggests the cells were growing at similar rates throughout the experiment. Are these maximal growth rates? If the flux of phosphate were increased, would the growth rates increase? Whether the cells are truly P limited is important, as it has implications on how to interpret the 65% increase in cellular P quota achieved at the same growth rate. It is tempting to speculate that this accumulation of poly-P under OA conditions is mandatory (for pH homeostasis), regardless of dissolved P availability, and I might expect that based on the data presented here, but I don't believe this has been demonstrated. If this accumulation of elevated POP:POC is mandatory under OA conditions, then under "growth rate limiting" steady state conditions (where growth rate = steady state uptake divided by the required POP and uptake depends on $[PO_4^{3-}]$) one would expect a lower growth rate at a limiting concentration of $[PO_4^{3-}]$ (or SRP if you will) under OA conditions.

Fig 1A	rate (mol substrate per cell per hour), estimated from Fig 1					
Ambient C fixation	1.43E-13					
Acidified C fixation	1.19E-13					
Ambient N fixation	7.56E-15					
Acidified N fixation	5.32E-15					
Fig 1B	(mol per cell)					
Ambient POC	8.00E-12					
Acidified POC	6.19E-12					
Ambient PON	1.16E-12					
Acidified PON	9.58E-13					
Ambient POP	2.43E-14					
Acidified POP	3.11E-14					
from Fig1A&1B	specific rates					
Ambient C fixation	1.79E-02					
Acidified C fixation	1.92E-02					
Ambient N fixation	6.49E-03					
Acidified N fixation	5.55E-03					
Calculated differences						
AmbVSAcid C fix	1.07E+00 (7% increase)					
AmbVSAcid N fix	8.56E-01 (14% decrease)					
POP:POC (mol:mol)			<<--- therefore, POC:POP (compare to Supp Table 1)			
Ambient	3.04E-03		3.29E+02			
Acidified	5.02E-03		1.99E+02			
% change in POP:POC	65%		(much greater than 35%)			
Chlorophyll						
Ambient chla	8.18E-13					
Acidified chla	5.70E-13					
Normalized						
(gram chla: mol C)						
Ambient chla:C	1.02E-01					
Acidified chla:C	9.21E-02					
% change in chla:C	10%					

Since I have different interpretation of the lab experiments, and I am not certain whether the data from Fig 1A/B or from Supp Table 1 were used as input for the biogeochemical models, I do not have any comment at this time.

Some very minor comments:

I do not understand the significant figures in Supplemental Table 2 (cells in outflow); I had assumed this was calculated as the product of cells per mL and mL per day.

Line 178: Consider using decreased rather than reduction of, since reduction has a distinct chemical meaning, esp in the context of NADP/NADPH.

Reviewer #2 (Remarks to the Author):

The paper by Zhang et al. presents laboratory, modeling and field experiment results demonstrating that under phosphate limitation, ocean acidification would lead to greater reduction of carbon and nitrogen metabolism in *Trichodesmium* compared to non-acidified condition. Interestingly, despite phosphate limitation, the phosphorus cell content was higher in the acidified treatment and polyphosphate was found to be the pool likely responsible for this increase. The authors present evidence suggesting that the increased polyphosphate may help *Trichodesmium* maintain cytosolic pH homeostasis under acidification. Considering the importance of this N₂-fixing cyanobacterium in global ocean nitrogen cycling, particularly in providing a bioavailable source of nitrogen in the N-limited ocean, the findings of this paper are important. The paper is extremely well written and organized, the figures are polished. The methods are well detailed, which will help other groups replicate those experiments. The introduction and results and discussion are to the point. It was a real pleasure reading this manuscript. The authors present a beautiful dataset, encompassing the analysis of stocks, fluxes, and gene expression, with laboratory experiments under steady-state phosphate limitation that are ground thruted and which results are being used in a predictive model: this is a perfect example of where our field needs to go. I usually write long reviews, but I do not have many questions here.

My main criticism with this work is that the role of bioavailable DOP is completely ignored. I understand that the goal of this study is to test the effect of ocean acidification on phosphate limited *Trichodesmium*, but considering the recognized importance of DOP as a source of phosphate in phosphate limited environments (reviewed in Duhamel et al. 2021), including for *Trichodesmium* (e.g., Mulholland et al. 2002; Orchard et al. 2010), this should be discussed. This is particularly important when discussing the model results since N₂ fixation may not be reduced as dramatically if *Trichodesmium* can utilize the DOP pool.

I am also puzzled that the laboratory experiments did not include a P-replete reference. For example, in the abstract, L30-31: "Consequently...": how can you conclude that the negative effect of acidification was amplified under P-limiting conditions without a P-replete control?

Minor comments:

Title: I suggest using the term phosphate instead of phosphorus

L28-29: "Accumulation of polyP resulted in...": this should be rephrased because this is speculative.

L73: Since the methods come after the result and discussion section, please define what are the "ambient or acidified" conditions.

L81: Replace "demonstrating" by "suggesting" since the ratio is not a demonstration of P-limitation which is a physiological state. Perhaps the gene transcription data are a better demonstration.

L117: Replace "*Dunaliella saina*" by "*Dunaliella salina*"

L188-189: I would argue that POP and polyP can be measured in *Trichodesmium* since they can be hand-picked and rinsed. That said we know that it would be difficult to remove their bacterial epibionts. Still, they can be separated from the "natural phytoplankton assemblages".

L192-194: I find this sentence confusing.

L426: Replace "settling" by "setting"

L501: after "being" treated with

Duhamel, S., J. M. Diaz, J. C. Adams, K. Djaoudi, V. Steck, and E. M. Waggoner. "Phosphorus as an Integral Component of Global Marine Biogeochemistry." *Nature Geoscience* 14, no. 6 (June 1, 2021): 359–68. <https://doi.org/10.1038/s41561-021-00755-8>.

Mulholland, M.R., S. Floge, E. J. Carpenter, and D. G. Capone. "Phosphorus Dynamics in Cultures and Natural Populations of *Trichodesmium* Spp." *Marine Ecology - Progress Series* 239 (2002): 45–55.

Orchard, E. D., J. W. Ammerman, M. W. Lomas, and S. T. Dyhrman. "Dissolved Inorganic and Organic Phosphorus Uptake in *Trichodesmium* and the Microbial Community: The Importance of Phosphorus Ester in the Sargasso Sea." *Limnology and Oceanography* 55, no. 3 (May 2010): 1390–99. <https://doi.org/10.4319/lo.2010.55.3.1390>.

Reviewer #3 (Remarks to the Author):

The authors present data from a series of laboratory and field experiments which evidence an enhanced negative effect of ocean acidification on N₂ fixation for the important cyanobacterial diazotroph *Trichodesmium*. Moreover, a range of data is collected across these experiments which is used to construct a suggested physiological mechanism. The authors finally perform a series of offline calculations based on prior numerical modelling, to estimate a potential magnitude of the effect indicated for overall oceanic N₂ fixation at the end of the century.

The manuscript is generally well written and presented. Interesting results are presented which contain convincing evidence of an potential important interaction between ocean acidification and P limitation in controlling N₂ fixation in *Trichodesmium* and some plausible, although in places speculative, physiological underpinning proposed. Although I appreciated the overall effort to provide a comprehensive study incorporating laboratory, field and modelling methods, I found the final modelling component to be the weakest. Moreover I would actually suggest that, irrespective of the considerable caveats with the modelling (see below and noting that many of these are identified by the authors themselves), this ends up adding little of substance to an otherwise high quality piece of work.

I am therefore supportive of the publication of the work in some form. However, I below indicate a number of places where I would like to see some revision and, as indicated above, I would suggest the authors consider whether the modelling calculations actually have value or rather, as the authors indicate, this global scale extrapolation is better left to a more comprehensive modelling study.

Major comments:

The arguments around the detailed cellular mechanisms underpinning the observed physiological responses are somewhat speculative. I would have liked to see some acknowledgement of this and potentially further consideration of other plausible mechanisms. Some detailed on this are outlined in the specific comments below.

The calculations based on prior modelling were difficult to assess, as the explanation of how these calculations were undertaken was, at least for this reviewer, a bit difficult to follow. Irrespectively, given the significant caveats, including those highlighted by the

authors themselves and others (see below), I didn't feel this section of the manuscript added real value to the overall conclusions.

Additional specific comments:

Line 55: potential worth noting that the supply of N will also vary in the future alongside that of P. This is one further reason why I felt the calculations in the modelling section were probably too simplistic to add much of value.

Line 77: '...oceans which Trichodesmium inhabits...'

Line 88: Not sure this fully demonstrates the limitation, rather it is consistent with limitation by P.

Lines 97-99: some comparison with P replete cultures would have been useful here.

Line 111: '...was not largely affected...'

Lines 123-126: are there other mechanisms for cytosolic pH homeostasis?

Line 137 suggest '...is(are) known...'

Line 142: There is some potential for conflation of the overall abundance of the NADP(H) pool and the turnover rates of this pool within this and subsequent sections. I think the authors are arguing that the NADP(H) pool size might be a constraint on reductant supply through this pool? Is there good evidence for this either in Trichodesmium or other organisms? If so please provide reference.

Lines 146-147: I wasn't convinced that low availability of NADP(H) would be a major constraint on chlorophyll synthesis as the latter will only be a minor component of the overall reductant requirement of the cell. Carbon and nitrogen fixation will both be larger ongoing requirements and I would speculate it is more likely that any limitation in reductant supply (although see previous point) would directly influence C and / or N fixation, with subsequent influences on chlorophyll (and PSI transcription) then potentially a result of cellular acclimation to any different energetic/reductant balance?

Line 166: A similar argument to above. Overall ATP use will more likely be dominated by other cellular processes (C fixation, N fixation). I am happy to be rebutted on this and previous point, e.g. authors might like to calculate/estimate potential ATP usage in maintenance of cytosolic pH homeostasis and overall NADP(H) use in chlorophyll synthesis (e.g. compare growth rate x chl per cell x NADPH per chl versus growth rate x N per cell x NADPH per N fixed), but ultimately I suspect the mechanisms behind the observed results are more complex than those outlined.

Lines 180-181: see further comment below.

Line 189: '...specifically targeting the Trichodesmium populations...'

Line 193: suggest '... expression potentially related to the need of more...'

Lines 196-213: It was unclear to me how this calculation was being performed. As mentioned above, I would suggest the whole section can be removed without much being lost, as the overall quantitative result is (presumably) highly uncertain. If retained then the explanation should be clarified.

Line 236: 'chemostat'

Line 342: '...after being kept in a -80...'

Line 372: 'bioinformatics'

Lines 421-423: The phosphate amendments don't appear to be explicitly mentioned in the main text? Lines 180-181 should be edited to more fully describe the evidence for P limitation of the experimental locations.

Line 444: 'At station OA-5, a Trichodesmium bloom...'

Lines 452-454: Could the different method of carbonate system manipulation have any influence on the results?

Line 501: '...after being treated...'

Line 599: repeated reference (see Ref 65, line 818)

Lines 615-621: As indicated by the authors, the CESM model does not explicitly represent the drivers of N₂ fixation being investigated in the rest of the manuscript. It is therefore not clear that use of this model output as the basis for subsequent calculations is valid as a mechanism to extrapolate findings to global scale or end of century. I also note the further caveats mentioned on lines 635-636 and lines 598-599.

Line 903: why no errorbars for the transcriptomic data?

Manuscript # NCOMMS-21-46616 by Zhang et al.

We thank the three Reviewers for their constructive comments on our manuscript. We have responded to these below in blue (Note: the line numbers mentioned in responses below correspond to those in the revised Word file with track changes).

Reviewer #1

The authors have investigated the role of OA on cellular phosphorus dynamics under steady state conditions where the inorganic P supply is low. I commend the authors for their significant effort to achieve culture conditions with a steady state supply of inorganic phosphorus, compared to the more trodden path whereby cells are grown under semi-continuous or starvation conditions. To me, the authors' data suggests an enhanced cellular P quota much greater than they have reported (described below), and the prospect of poly-P accumulation is very interesting and deserves further exploration. I have been impressed with the authors' scholarship over the last 15 years, but I have reservations on some of the interpretations presented here. I hope the authors will find these comments useful and constructive towards an improved manuscript.

We are thankful to the Reviewer for appreciating our efforts in carrying out this study using chemostat cultures to achieve steady-state P-limited conditions. We also thank the Reviewer for the insightful and constructive comments that have much improved the manuscript.

The authors report decreases in rates of C and N fixation due to OA conditions, and further report that the cellular PON and POC contents had also declined under OA compared to ambient conditions, "in accordance with the decreased fixation rates". I interpret their data differently, considering the biochemical relevance of the 'specific' rates. Comparing the appropriately normalized rates of C and N fixation to the cellular C and cellular N, respectively, suggests C and N fixation ~7% higher and ~14% lower under OA conditions (see attached table). I cannot help but wonder whether this 14% decrease is statistically significant. Similarly I estimate the C-normalized chlorophyll content of the smaller cells grown under OA conditions to be only 10% lower than that of ambient cells. In these regards, the emphasis and even the title of the manuscript may be reconsidered. If there is a silver lining here, it is that the POP:POC (ie cellular P quota) is actually 65% greater under OA conditions compared to ambient conditions (Note that Supplemental Table 1 is not consistent with data in Figs 1A and 1B; the reported acidified POC:POP appears to have been incorrectly calculated as the value from Figs 1A and 1B suggest a value of about 200:1 rather than 293:1 (mol:mol)).

We thank the Reviewer for this comment. The primary purpose and focus of our laboratory experiments were to delving into the underlying mechanisms of how *Trichodesmium* responds to OA under P-limited conditions. To this end, we cultured the diazotroph at a submaximal growth rate under highly controlled, P-limited conditions using chemostats, which is critical to

quantitative analysis and mechanistic understanding of the responses of cellular processes and activity. Therefore, C and N₂ fixation rates, POC, PON and POP, as well as Chl_a, ATP and NADP(H) contents on a per-cell basis would help to correlate these physiological and biochemical measurements to gene expression levels, thereby allowing us to explore the underlying mechanisms of cellular responses. As shown in the manuscript, we found that under acidified conditions, on a per-cell basis, the rate of N₂ fixation decreased by 30% ($p = 0.024$, Fig. 1a) and POP increased by 28% ($p < 0.001$, Fig. 1b; increased by 65% if normalized to cellular C, as the Reviewer suggested).

Nevertheless, we very much appreciate the Reviewer's comments that have inspired us to extrapolate our results to provide ecological and biogeochemical implications. As our study was to understand the effects of OA on *Trichodesmium* under P limitation and accordingly the chemostat experiments were set up to compare the diazotroph's response to different CO₂/pH levels given the same PO₄³⁻ supply, we think that from a biogeochemical prospect it would be more relevant to normalize C and N₂ fixation rates to cellular P quota rather than C or N quota. The calculations show that POP-normalized C and N₂ fixation rates both decreased (36% and 45 % lower, respectively) significantly under acidified conditions ($p = 0.004$ and 0.002 , respectively), suggesting elevated P demands for C and N₂ fixation (Lines 118-120). In addition, as the Reviewer pointed out, the significant increase of cellular P quota in response to OA also implies an enhanced cellular P requirement (**Note:** we thank the Reviewer for pointing out the inconsistency of the data in Fig. 1b and in Supplementary Table 2. The POP and POC data in Fig. 1b are correct, but the POC:POP was incorrectly calculated. We have now recalculated POC:POP which are 332 ± 4 and 202 ± 14 , respectively, under ambient and acidified conditions), and a description of the biogeochemical significance of changes in PON:POP is noted in the concluding section of the manuscript (Lines 359-365).

We have now added cellular POP-normalized C and N₂ fixation rates in Supplementary Table 2, and revised the manuscript accordingly (Lines 118-120).

If I understand the chemostat data properly (Supplemental Figure 1) the cells in both treatments were growing at the same rate. That the dilution rates of the chemostats under both conditions were set to 0.3 per day (0.3 litres per litre per day) and the cells were not washed out suggests the cells were growing at similar rates throughout the experiment.

Yes, cells in both ambient and acidified treatments were growing at a same rate in the chemostats. In our chemostats, the inflow and outflow rates were 0.3 L day⁻¹, and the constant culture volume was maintained at 1.5 L. Therefore, the dilution rates (D) were 0.2 day⁻¹ (i.e., inflow rate/constant volume). In a chemostat system, the change in biomass dX over an infinitely small-time interval (dt) can be expressed as

$$dX/dt = \mu X - DX = X(\mu - D)$$

where X is cell density (cell L^{-1}), μ is specific growth rate, and D is dilution rate (Rhee 1989; Borchard et al. 2011).

After reaching dynamic equilibrium (steady-state), the cell density remains constant and $dX/dt = 0$, and thus the specific growth rate equals the dilution rate ($\mu = D$). Therefore, in our study, cells under both ambient and acidified conditions were growing at the same rate of 0.2 day^{-1} .

Are these maximal growth rates?

In a nutrient-limited chemostat, the enforced growth rate is always lower than the potential maximal growth rate of the investigated species once a steady state is accomplished. Given the same temperature and irradiation intensity as in this study, the maximum specific growth rate of *Trichodesmium* can reach $\sim 0.5 \text{ day}^{-1}$ under nutrient replete conditions (e.g., Zhang et al. 2019).

If the flux of phosphate were increased, would the growth rates increase?

The flux of phosphate = inflow rate \times $S_r = D \times V \times S_r$

where D is the dilution rate, V is the constant volume, and S_r is the concentration of phosphate in medium supply.

If the flux of phosphate is increased by increasing D to a D_{new} , this will change the status of the system from steady state to non-steady-state. A new steady-state will be reached after the cells adjust themselves to a new specific growth rate μ_{new} , which equals to D_{new} . If the flux of phosphate is increased by increasing S_r , the growth rates will not change and will remain equal to the dilution rate (D).

Whether the cells are truly P limited is important, as it has implications on how to interpret the 65% increase in cellular P quota achieved at the same growth rate.

In our culture systems, when reaching steady-state, (i) the concentrations of phosphate in the chemostats were only $\sim 10\text{-}20 \text{ nM}$ under either ambient or acidified conditions, which were much lower than that in the supply medium (i.e., $1.2 \text{ }\mu\text{M}$), suggesting quick and almost complete consumption of the phosphate supplied; and (ii) the activity of alkaline phosphatase of *Trichodesmium*, a molecular diagnostic of P stress in *Trichodesmium*, increased significantly compared to that under P-replete conditions (7.3-fold, Supplementary Figure 2a). In addition, the observed high PON:POP ratio (48:1) was close to those of *Trichodesmium* in P-limited North Atlantic. All these results suggested that *Trichodesmium* in our chemostat cultures were P-limited.

It is tempting to speculate that this accumulation of poly-P under OA conditions is mandatory (for pH homeostasis), regardless of dissolved P availability, and I might expect that based on the data presented here, but I don't believe this has been demonstrated. If this accumulation of elevated POP:POC is mandatory under OA conditions, then under “growth rate limiting” steady state conditions (where growth rate = steady state uptake divided by the required POP and uptake depends on $[\text{PO}_4^{3-}]$) one would expect a lower growth rate at a limiting concentration of $[\text{PO}_4^{3-}]$ (or SRP if you will) under OA conditions.

The physiological functions of polyP in cyanobacteria including *Trichodesmium* still remain to be fully understood. Taking into account our previous findings that OA can affect intracellular pH homeostasis of *Trichodesmium* (Hong et al. 2017) and the reported pH-homeostatic function for polyP in some microorganisms (e.g., McGrath and Quinn 2000), we speculate that the accumulation of polyP under OA is plausibly for cytosolic pH homeostasis in *Trichodesmium*.

As illustrated in the conceptual figure below, in an “ideal” steady-state PO_4^{3-} -limited system (Case I), for a given concentration of phosphate (i.e., $[\text{PO}_4^{3-}]_A$), a lower growth rate is expected under acidified conditions. However, realistically, such an “ideal” system is difficult to be experimentally realized. In a chemostat system (Case II), such as the one in our study that uses a same dilution rate (i.e., growth rate) for both ambient and acidified treatments, a higher concentration of phosphate is needed for cells grown under acidified conditions (i.e., $[\text{PO}_4^{3-}]_A > [\text{PO}_4^{3-}]_B$). But, in chemostats, because of quick and almost complete consumption of the phosphate supplied, both $[\text{PO}_4^{3-}]_A$ and $[\text{PO}_4^{3-}]_B$ are very low. In the case of our study, $[\text{PO}_4^{3-}]_A$ and $[\text{PO}_4^{3-}]_B$ were $< 20 \text{ nM}$ and the difference between them could not be well differentiated by the analytical method we used.

Since I have different interpretation of the lab experiments, and I am not certain whether the data from Fig 1A/B or from Supp Table 1 were used as input for the biogeochemical models, I do not

have any comment at this time.

We used changes in C biomass (i.e., POC:POP) and POC-normalized N₂ fixation rate in response to ocean acidification and/or P limitation as the input for model calculations. To clarify these, we have now thoroughly revised the “Model estimations” section in Methods. Please see Lines 794-893 and the data in Supplementary Table 2.

Some very minor comments:

I do not understand the significant figures in Supplemental Table 2 (cells in outflow); I had assumed this was calculated as the product of cells per mL and mL per day.

We thank the Reviewer for pointing this out. Yes, cells in outflow (cell d⁻¹) = cell density (cell mL⁻¹) × flow rate (L d⁻¹), where the cell density was the average of three measurements. In the original calculation, we kept too many significant digits/figures of the cell density. We have now corrected the significant figures and redone the calculations (new Supplementary Table 3).

Line 178: Consider using decreased rather than reduction of, since reduction has a distinct chemical meaning, esp in the context of NADP/NADPH.

Thanks for the suggestion. “reduction of” has now been replaced by “decreased” (Line 218).

References

- Borchard, C., Borges, A.V., Händel, N., & Engel, A. Biogeochemical response of *Emiliania huxleyi* (PML B92/11) to elevated CO₂ and temperature under phosphorous limitation: a chemostat study. *J. Exp. Mar. Biol. Ecol.* **410**, 61-71 (2011)
- Hong, H. *et al.* The complex effects of ocean acidification on the prominent N₂-fixing cyanobacterium *Trichodesmium*. *Science* **356**, 527-531 (2017).
- McGrath, J. W. & Quinn, J. P. Intracellular accumulation of polyphosphate by the yeast *Candida humicola* G-1 in response to acid pH. *Appl. Environ. Microbiol.* **66**, 4068-4073 (2000).
- Rhee, G.Y. Continuous culture algal bioassays for organic pollutants in aquatic ecosystems. *Hydrobiologia* **188**, 247-258 (1989)
- Zhang, F. T. *et al.* Proteomic responses to ocean acidification of the marine diazotroph *Trichodesmium* under iron-replete and iron-limited conditions. *Photosynth. Res.* **142**, 17-34 (2019).

Reviewer #2

The paper by Zhang et al. presents laboratory, modeling and field experiment results demonstrating that under phosphate limitation, ocean acidification would lead to greater reduction of carbon and nitrogen metabolism in *Trichodesmium* compared to non-acidified condition. Interestingly, despite phosphate limitation, the phosphorus cell content was higher in the acidified treatment and polyphosphate was found to be the pool likely responsible for this increase. The authors present evidence suggesting that the increased polyphosphate may help *Trichodesmium* maintain cytosolic pH homeostasis under acidification. Considering the importance of this N₂-fixing cyanobacterium in global ocean nitrogen cycling, particularly in providing a bioavailable source of nitrogen in the N-limited ocean, the findings of this paper are important.

The paper is extremely well written and organized, the figures are polished. The methods are well detailed, which will help other groups replicate those experiments. The introduction and results and discussion are to the point. It was a real pleasure reading this manuscript. The authors present a beautiful dataset, encompassing the analysis of stocks, fluxes, and gene expression, with laboratory experiments under steady-state phosphate limitation that are ground thruted and which results are being used in a predictive model: this is a perfect example of where our field needs to go. I usually write long reviews, but I do not have many questions here.

We very much appreciate the positive comments of the Reviewer on our study.

My main criticism with this work is that the role of bioavailable DOP is completely ignored. I understand that the goal of this study is to test the effect of ocean acidification on phosphate limited *Trichodesmium*, but considering the recognized importance of DOP as a source of phosphate in phosphate limited environments (reviewed in Duhamel et al. 2021), including for *Trichodesmium* (e.g., Mulholland et al. 2002; Orchard et al. 2010), this should be discussed. This is particularly important when discussing the model results since N₂ fixation may not be reduced as dramatically if *Trichodesmium* can utilize the DOP pool.

We fully agree with the Reviewer that DOP can represent a significant P source for *Trichodesmium* growth in surface waters of oligotrophic oceans. Following this comment, we have now added a new figure to show that the activity of alkaline phosphatase, a molecular diagnostic of P stress in *Trichodesmium*, observed when the chemostat cultures reached steady-state conditions was significantly higher than that of P-sufficient *Trichodesmium* (Supplementary Figure 2a). This not only suggests that *Trichodesmium* in our chemostats were P-limited, but also indicates an increased utilization of DOP, despite the fact that DOP was estimated to provide a very limited portion of the cellular P necessary for *Trichodesmium* growth (on average 4.5%, new Supplementary Table 3) in our chemostat cultures.

All these have now been added in the manuscript (Lines 112-114), and a discussion (Lines 132-146) on the importance of DOP as a P source for *Trichodesmium* under phosphate-limited conditions is reproduced below:

“Aside from PO_4^{3-} , *Trichodesmium* spp. are able to utilize phosphite and dissolved organic phosphorus (DOP), including the monophosphate esters (C-O-P bond) and phosphonates (C-P bond), to cope with P limitation (Mulholland et al. 2002; Orchard et al. 2010). In our culture media, DOP was not the added P source and was estimated to support only a few percent of POP at the most (Supplementary Table 3). We found that gene transcriptions of alkaline phosphatases (e.g., *phoX2* and *phoA*)²⁵ were not significantly affected while those involved in phosphonate (and/or phosphite) utilization (i.e., *phnD1/ptxA*, *phnC1/ptxB*, *phnE1/ptxC*, *ptxD*, and *Tery_2885*) were markedly up-regulated by OA (Supplementary Figure 2b)”.

In addition, we have also added in the concluding section that the effects of OA on DOP utilization and thus N_2 fixation by *Trichodesmium* warrant further investigation (Lines 341-344). And we have also clarified in the modeling part that *Trichodesmium* can use both DIP and DOP for their growth in the model, and the Biogeochemical Elemental Cycling (BEC) model takes both DIP and DOP into account in calculating P limitation strength (Lines 852-853).

I am also puzzled that the laboratory experiments did not include a P-replete reference. For example, in the abstract, L30-31: “Consequently...”: how can you conclude that the negative effect of acidification was amplified under P-limiting conditions without a P-replete control?

We acknowledge the lack of a P-replete reference in the present study. However, in Zhang et al. (2019) we quantified the OA effect under P-replete conditions. Here we compared the magnitude of OA effects under P-limited conditions to that under P-replete conditions reported in Zhang et al. (2019), and found that it was much amplified. Therefore, we do agree with the Reviewer that this should be clarified in the manuscript, and have now rephrased this sentence in the abstract as “Consequently, the negative effects of acidification were amplified compared to that reported previously under P-replete conditions.” (Lines 31-32).

Minor comments:

Title: I suggest using the term phosphate instead of phosphorus

Thanks for the suggestion. Phosphorus has now been replaced by phosphate in the title.

L28-29: “Accumulation of polyP resulted in...”: this should be rephrased because this is speculative.

We agree. Rephrased as “Alongside the accumulation of polyP, decreased NADP(H):NAD(H) ratios and impaired chlorophyll synthesis and energy production were observed under acidified conditions.” (Lines 29-31)

L73: Since the methods come after the result and discussion section, please define what are the “ambient or acidified” conditions.

Thanks for the suggestion. We have now defined ambient and acidified conditions here (Lines 93-94).

L81: Replace “demonstrating” by “suggesting” since the ratio is not a demonstration of P-limitation which is a physiological state. Perhaps the gene transcription data are a better demonstration.

Revised (Lines 104-105).

L117: Replace “Dunaliella saina” by “Dunaliella salina”

Thanks for pointing this out. Corrected (Line 159).

L188-189: I would argue that POP and polyP can be measured in *Trichodesmium* since they can be hand-picked and rinsed. That said we know that it would be difficult to remove their bacterial epibionts. Still, they can be separated from the “natural phytoplankton assemblages”.

If *Trichodesmium* forms colonies, they can be relatively, easily hand-picked and separated from the natural phytoplankton assemblages. However, in six out of the seven field experiments we conducted in the northern South China Sea, the natural *Trichodesmium* populations collected from the surface seawaters were present as free trichomes, not colonies. Therefore, it was very difficult for us to separate them from the assemblages.

L192-194: I find this sentence confusing.

The sentence has been rephrased (Lines 296-299).

L426: Replace “settling” by “setting”

Revised (Line 583).

L501: after “being” treated with

“being” added (Line 666).

References

Zhang, F. T. *et al.* Proteomic responses to ocean acidification of the marine diazotroph *Trichodesmium* under iron-replete and iron-limited conditions. *Photosynth. Res.* **142**, 17-34 (2019).

Reviewer #3

The authors present data from a series of laboratory and field experiments which evidence an enhanced negative effect of ocean acidification on N₂ fixation for the important cyanobacterial diazotroph *Trichodesmium*. Moreover, a range of data is collected across these experiments which is used to construct a suggested physiological mechanism. The authors finally perform a series of offline calculations based on prior numerical modelling, to estimate a potential magnitude of the effect indicated for overall oceanic N₂ fixation at the end of the century.

The manuscript is generally well written and presented. Interesting results are presented which contain convincing evidence of an potential important interaction between ocean acidification and P limitation in controlling N₂ fixation in *Trichodesmium* and some plausible, although in places speculative, physiological underpinning proposed. Although I appreciated the overall effort to provide a comprehensive study incorporating laboratory, field and modelling methods, I found the final modelling component to be the weakest. Moreover I would actually suggest that, irrespective of the considerable caveats with the modelling (see below and noting that many of these are identified by the authors themselves), this ends up adding little of substance to an otherwise high quality piece of work.

I am therefore supportive of the publication of the work in some form. However, I below indicate a number of places where I would like to see some revision and, as indicated above, I would suggest the authors consider whether the modelling calculations actually have value or rather, as the authors indicate, this global scale extrapolation is better left to a more comprehensive modelling study.

We thank the Reviewer for the positive comments and the time the Reviewer has taken to suggest improvements. In particular, we appreciate the suggestion on considering whether the modelling calculations are actually of value. The impacts of ocean acidification and consequent changes in marine N₂ fixation are certainly important topics that are of broad interest to ecologists, biogeochemists, oceanographers and climate change scientists and are thus worthy of comprehensive modelling studies.

However, a more comprehensive modelling study of the effects of acidification and P limitation on marine N₂ fixation would require:

(i) simulations with explicit representations of various N₂-fixing functional groups in CESM, which are in need of further experimental results to provide key information for model developments. For example, it would involve key diazotrophic phylotypes, including not only *Trichodesmium* but also the unicellular symbiont UCYN-A, the free-living unicellular *Crocospaera* (UCYN-B) and the diatom symbionts. However, as in most Earth system models, there is only one diazotrophic functional group representing all N₂ fixers in the Biogeochemical Elemental Cycling Model.

(ii) more importantly, a comprehensive understanding of the impacts of global change on marine N₂ fixation. However, N₂ fixation by different diazotrophic groups is subject to a battery of biotic and abiotic factors (e.g., OA, Fe, P, temperature, and grazing pressure) whose effects are very complex (individually, additively, antagonistically, or synergistically) and still remain to be fully understood.

(iii) ideally, a more comprehensive simulation of OA impacts on all phytoplankton groups, which is beyond the boundaries of current knowledge.

Therefore, without aforementioned further, comprehensive observational and experimental results, explicit model representations and dynamic simulations of the effects of OA and P limitation on marine N₂ fixation at the moment would have to base on too many assumptions, which would lead to greater uncertainties. For that reason, in the present study, the main purpose of the modelling component is to illustrate the implications of our experimental findings for the large-scale effects of OA coupled with P limitation on N₂ fixation by *Trichodesmium*. We consider uncertainties with such model calculations are less compared to a comprehensive modelling simulation and prediction would cause. We hope the Reviewer would agree with us that our model calculations can provide a first-order estimate of the spatial pattern of the impacts and an estimate of the large-scale effects on N₂ fixation of *Trichodesmium*. Particularly this will allow the audience who do not have much background in ocean biogeochemistry to better understand the ecological and biogeochemical implications of our study.

We think the model estimate in our study is a good starting point, which suggests that N₂ fixation in the Indian and west Pacific oceans are strongly affected. We consider this reasonable as it agrees well with previous estimated patterns of acidification and P limitation (e.g., Feely et al. 2009; Moore et al. 2013), and believe that the spatial pattern shown in this study can be valuable for planning future field campaigns.

Major comments:

The arguments around the detailed cellular mechanisms underpinning the observed physiological responses are somewhat speculative. I would have liked to see some acknowledgement of this and potentially further consideration of other plausible mechanisms. Some detailed on this are outlined in the specific comments below.

We thank the Reviewer for the detailed comments that have much improved the manuscript. Please see below for details.

The calculations based on prior modelling were difficult to assess, as the explanation of how these calculations were undertaken was, at least for this reviewer, a bit difficult to follow. Irrespectively, given the significant caveats, including those highlighted by the authors themselves and others (see below), I didn't feel this section of the manuscript added real value to

the overall conclusions.

We refer to our responses to the Reviewer comments above for why we feel that this should be included in the manuscript. We have now thoroughly revised the manuscript to provide more explanation of how the model calculations were undertaken. Please see Methods (Lines 748-893).

Additional specific comments:

Line 55: potential worth noting that the supply of N will also vary in the future alongside that of P. This is one further reason why I felt the calculations in the modelling section were probably too simplistic to add much of value.

Following the Reviewer's suggestion, we have now noted in the revised manuscript that the supply of N will also change alongside that of P (Line 64). And we refer to our responses to the Reviewer comments above for why we include this modelling section in the manuscript.

Line 77: '...oceans which *Trichodesmium* inhabits...'

Corrected (Line 105).

Line 88: Not sure this fully demonstrates the limitation, rather it is consistent with limitation by P.

Reviewer 2 had a similar comment. We have now revised this section (Lines 102-105).

Lines 97-99: some comparison with P replete cultures would have been useful here.

Thanks for the suggestions. We have now added a new figure (Supplementary Figure 2a) to show that the activity of alkaline phosphatase, observed when the chemostat cultures reached steady-state conditions, was significantly higher than that of P-sufficient *Trichodesmium* (Lines 102-104).

Line 111: '...was not largely affected...'

Corrected (Line 153).

Lines 123-126: are there other mechanisms for cytosolic pH homeostasis?

The cellular mechanisms for cytosolic pH homeostasis in marine phytoplankton generally include H⁺ buffering, metabolic H⁺ consumption and production and transmembrane H⁺

transport (Taylor et al. 2012; Bunse et al. 2016). We have now added this information (Lines 163-165).

Line 137 suggest ‘...is(are) known...’

Corrected (Line 193).

Line 142: There is some potential for conflation of the overall abundance of the NADP(H) pool and the turnover rates of this pool within this and subsequent sections. I think the authors are arguing that the NADP(H) pool size might be a constraint on reductant supply through this pool? Is there good evidence for this either in *Trichodesmium* or other organisms? If so please provide reference.

We thank the Reviewer for this comment. Actually, we did not try to argue that NADP(H) pool size might be a constraint on reductant supply. We have now clarified this by focusing the discussion on OA effects on the changes in the NADP(H):NAD(H) ratio (Lines 197-222). To the best of our knowledge, the physiological functions of the intracellular balance of NADP(H) and NAD(H) in *Trichodesmium* is unknown. However, the NADP(H):NAD(H) ratio is known to play an important role in regulating plant physiology, as the functions of NAD(H) and NADP(H) are significant and distinct (e.g., Kawai and Murata 2008; Takahashi et al. 2009). For instance, in NAD kinase-deficient *Arabidopsis*, the NADP(H):NAD(H) ratio was significantly reduced and the NADPH-dependent biosynthetic pathways such as carbon fixation and chlorophyll synthesis were hampered (Chai et al. 2005; Takahashi et al. 2009). It has also been reported that the NAD kinase-deficient cyanobacterium *Synechocystis* sp. PCC6803 had a lower NADP(H) but a higher NAD(H) content, and more importantly a lower NADP(H):NAD(H) ratio (Ishikawa et al. 2016). Similarly, in our study, in line with the reduced gene transcription of NAD kinase and the decreased NADP(H):NAD(H) ratio (Fig. 2), we observed a considerable decrease in carbon fixation rate (Fig. 1a) and a lower cellular Chl_a content (Fig. 2c) under acidified conditions.

Lines 146-147: I wasn't convinced that low availability of NADP(H) would be a major constraint on chlorophyll synthesis as the latter will only be a minor component of the overall reductant requirement of the cell. Carbon and nitrogen fixation will both be larger ongoing requirements and I would speculate it is more likely that any limitation in reductant supply (although see previous point) would directly influence C and / or N fixation, with subsequent influences on chlorophyll (and PSI transcription) then potentially a result of cellular acclimation to any different energetic/reductant balance?

Line 166: A similar argument to above. Overall ATP use will more likely be dominated by other cellular processes (C fixation, N fixation). I am happy to be rebutted on this and previous point, e.g. authors might like to calculate/estimate potential ATP usage in maintenance of cytosolic pH homeostasis and overall NADP(H) use in chlorophyll synthesis (e.g. compare growth rate x chl

per cell x NADPH per chl versus growth rate x N per cell x NADPH per N fixed), but ultimately I suspect the mechanisms behind the observed results are more complex than those outlined.

As the two comments above are related, here we respond them together.

We fully agree with the Reviewer that (i) the NADP(H) requirements for both C fixation and N₂ fixation (actually in the subsequent GS/GOGAT reaction) are much larger than those for chlorophyll synthesis; (ii) the ATP requirements for maintaining cytosolic pH homeostasis, which are difficult to estimate quantitatively, could also be less than those for C and N₂ fixation; and (iii) the mechanisms behind the observed changes in C and N₂ fixation rate, chlorophyll content, and cytosolic pH homeostasis and the associated variation in the usages of reductant and energy are very complex.

The physiological functions of NAD kinase and the intracellular balance of NADP(H) and NAD(H) in cyanobacteria are poorly understood. However, it has been shown in *Arabidopsis* that NADP(H) concentrations and NADP(H):NAD(H) ratios are proportional to NAD kinase activity (e.g., Takahashi et al. 2009). Since the reducing energy of NADPH is used for biological processes such as chlorophyll synthesis and CO₂ fixation, NAD kinase activity affects both chlorophyll synthesis and CO₂ fixation (Takahashi et al. 2009). In particular, it was found that chlorophyll content was reduced in the NAD kinase-deficient strain of *Arabidopsis* (Chai et al. 2005; Takahashi et al. 2009). Moreover, Chai et al. (2005) has reported that NADPH:protochlorophyllide (Pchl_{id}) and Mg-protoIV, which required NADPH as reducing energy to be converted to the next step intermediates during chlorophyll synthesis was found to accumulate in NAD kinase-mutant strain that contained lower NADP(H) concentrations and NADP(H):NAD(H) ratios.

Therefore, based on these findings, we speculate that in P-limited *Trichodesmium*, in line with the down-regulation of NAD kinas gene transcription under acidified conditions, an increased in NAD(H) and a decrease in NADP(H) concentrations and hence a decrease in NADP(H):NAD(H) ratio, directly or indirectly, affected C and N₂ fixation, as well as other cellular processes such as chlorophyll synthesis. In addition, the shortage of ATP supply should have also been responsible for the decreased C and N₂ fixation rates. Moreover, the energy required for maintaining cytosolic pH homeostasis by plasma membrane transporters would become insufficient, and thus *T. erythraeum* would have relied on increased polyP likely for cytosolic pH homeostasis under acidified conditions. We have now revised the discussion accordingly (Lines 197-265).

Lines 180-181: see further comment below.

We refer to our responses to the Reviewer comments below.

Line 189: ‘...specifically targeting the *Trichodesmium* populations...’

Corrected (Line 294).

Line 193: suggest ‘... expression potentially related to the need of more...’

Thanks for the suggestion. Rephrased as suggested (Line 298).

Lines 196-213: It was unclear to me how this calculation was being performed. As mentioned above, I would suggest the whole section can be removed without much being lost, as the overall quantitative result is (presumably) highly uncertain. If retained then the explanation should be clarified.

We have now thoroughly revised the Methods to provide more details on how the calculation was performed (Lines 748-893). And we refer to our responses to the Reviewer comments above for why we include this modelling component in the manuscript.

Line 236: ‘chemostat’

Corrected (Line 367).

Line 342: ‘...after being kept in a -80...’

Corrected (Line 493).

Line 372: ‘bioinformatics’

Corrected (Line 523).

Lines 421-423: The phosphate amendments don’t appear to be explicitly mentioned in the main text? Lines 180-181 should be edited to more fully describe the evidence for P limitation of the experimental locations.

Thanks for the suggestion. We have now rephrased the text to describe the evidence of P limitation of the natural *Trichodesmium* populations in surface waters collected in our study region as demonstrated by the phosphate amendment experiments (Lines 277-280).

Line 444: ‘At station OA-5, a *Trichodesmium* bloom...’

Corrected (Line 603).

Lines 452-454: Could the different method of carbonate system manipulation have any influence on the results?

It has been shown in a previous study by the corresponding author that experimentally the different method of carbonate system manipulation does not affect the results (Shi et al. 2009).

Line 501: ‘...after being treated...’

Corrected (Line 666).

Line 599: repeated reference (see Ref 65, line 818)

Thanks for pointing this out. The repeated reference has now been removed.

Lines 615-621: As indicated by the authors, the CESM model does not explicitly represent the drivers of N₂ fixation being investigated in the rest of the manuscript. It is therefore not clear that use of this model output as the basis for subsequent calculations is valid as a mechanism to extrapolate findings to global scale or end of century. I also note the further caveats mentioned on lines 635-636 and lines 598-599.

To the best of our knowledge, individual diazotrophic groups are not represented explicitly in any Earth system models. CESM, as in other ESMs, has only one functional group representing all N₂ fixers. We therefore have to adopt offline calculations which can incorporate the estimated contributions of *Trichodesmium* to the total N₂ fixation. Instead of extrapolating the laboratory results to all N₂ fixers, our calculation provides a much more conservative estimate of the effects of acidification and P limitation on *Trichodesmium* N₂ fixation. It is possible that the contributions of various diazotrophic groups to the total N₂ fixation would change by the end of this century, as different groups may adapt to climate change differently. This is a common caveat of all ecological/biogeochemical models using the concept of functional groups, which cannot simulate adaptation of phytoplankton to climate change. We consider the current model estimate a starting point to initiate reconsiderations of global estimates of N₂ fixation, and plan on developing a more comprehensive model when more information is available.

Line 903: why no errorbars for the transcriptomic data?

The transcriptomic data were obtained with experimental samples of three biological replicates, and were analyzed using the DESeq2 R package for differentially expressed genes. The analysis gave Log₂ Fold change, lfcSE (standard error of the Log₂ Fold change), and *p*-value adjusted using the Benjamini and Hochberg’s approach for controlling the false discovery rate. To directly compare and present the results obtained with RT-qPCR analysis (Fig. 2b), Log₂ Fold change

was converted to % change by calculations (i.e., $[2^{\text{Log}_2 \text{Fold change}} - 1] \times 100$). We have now added all this information in Supplementary Table 4 (**Note:** in compiling the data for the new Supplementary Table 4, we realized that the asterisks denoted *p*-values of the transcriptomic data in the original Fig. 2b were without the Benjamini and Hochberg correction and were mistakenly used. We have now corrected by replacing them with the Benjamini and Hochberg approach adjusted *p*-values. Although changes in the transcription of a few genes are now insignificant, given that the transcriptomic data and RT-qPCR data corroborated each other and different genes of a same functional group showed the same response, this change has no impact on the conclusions as reported in the original manuscript). In addition, a new Supplementary Table 5 is now added to present the RT-qPCR data (i.e., mean \pm standard deviation with statistical analysis) and how % change of the data was calculated.

References

- Bunse, C. *et al.* Response of marine bacterioplankton pH homeostasis gene expression to elevated CO₂. *Nat. Clim. Change* **6**, 483-487 (2016).
- Chai, M. F. *et al.* NADK2, an *Arabidopsis* chloroplastic NAD kinase, plays a vital role in both chlorophyll synthesis and chloroplast protection. *Plant. Mol. Biol.* **59**, 553-564 (2005).
- Feely, R. A., Doney, S. C. & Cooley, S. R. Ocean acidification: present conditions and future changes in a high-CO₂ World. *Oceanography* **22**: 36-47 (2009).
- Ishikawa, Y. *et al.* Metabolomic analysis of NAD kinase-deficient mutants of the cyanobacterium *Synechocystis* sp. PCC 6803. *J. Plant Physiol.* **205**: 105-112 (2016).
- Kawai, S. & Murata, K. Structure and function of NAD kinase and NADP phosphatase: key enzymes that regulate the intracellular balance of NAD(H) and NADP(H). *Biosci. Biotechnol. Biochem.* **72**, 919-930 (2008).
- Moore, J. K., Lindsay, K., Doney, S. C., Long, M. C. & Misumi, K. Marine ecosystem dynamics and biogeochemical cycling in the community earth system model [CESM1(BGC)]: comparison of the 1990s with the 2090s under the RCP4.5 and RCP8.5 scenarios. *J. Climate* **26**: 9291-9312 (2013).
- Shi, D., Xu, Y. & Morel, F. M. M. Effects of the pH/pCO₂ control method on medium chemistry and phytoplankton growth. *Biogeosci.* **6**, 1199-1207 (2009).
- Takahashi, H. *et al.* Pleiotropic modulation of carbon and nitrogen metabolism in *Arabidopsis* plants overexpressing the *NAD kinase2* gene. *Plant Physiol.* **151**, 100-113 (2009).
- Taylor, A. R., Brownlie, C. & Wheeler, G. L. Proton channels in algae: reasons to be excited. *Trends Plant Sci.* **17**, 1360-1385 (2012).

REVIEWER COMMENTS

Reviewer #2

Please note, reviewer #2 was asked to check the responses to reviewer #1 after reviewer #3 had already checked the responses to reviewer #2

Comments on responses to reviewer 1. I refer to the page number in the rebuttal pdf document.

Page 1: I agree with reviewer 1 that normalizing rates of C and N fixation to cellular C and N, respectively, is an important result that should be presented in the paper. While the authors make a strong case as to why the POP-normalized rates are informative in the context of their experimental design, they provide different information and I would recommend presenting (including statistics) and discussing both ways of normalizing. Indeed, knowing that P-limited cells can vary their P content via a myriad of mechanisms, normalizing rates of C and N fixation to P does not tell us how OA affects the 'specific' rates which should ideally be C to C and N to N, notwithstanding their respective biases.

Page 2, 3 and 4: The authors answer several questions posed by the reviewer 1 but do not explain how they clarified in the manuscript. This should have been addressed more clearly. I found some of the relevant information in the revised manuscript. However, I think that additional information found in the responses to reviewer 1 should also be found in the manuscript. In particular, the justification as to why the P-limited acidified treatment was not showing lower growth rates as one might expect.

Page 5: My expertise with modeling is very limited so I would prefer that an expert weigh on this, but thank you for clarifying where to find the changes in the manuscript. L852 of the revised manuscript, you indicate using both DIP and DOP for *Trichodesmium*'s growth in the model. Could you provide details? Indeed, the labile fraction of DOP being unknown, I am wondering how this is taken into account. If the total concentration of both forms of P are considered in the "limiting factor calculations", I would expect that *Trichodesmium* would not be P limited. Speaking of which, could you explain what this "limiting factor calculations" is?

L95 of the manuscript: the term "dissolved reactive phosphorus" is unusual, why not using "soluble reactive phosphorus"? Are you meaning something different?

L202: I suggest replacing the word "significant" since it is a statistical term, and therefore inappropriate here. Perhaps "important" would be a better choice.

L219: I suggest removing "indeed".

L341: The bioavailability of DOP is important, but it is unclear to me how it was considered in the model (see my comment above). It is important because the way the abstract is phrased, one may interpret literally that *Trichodesmium* N₂ fixation will decrease by 38% due to ocean acidification in DIP limited environments. However, we know that *Trichodesmium* can utilize DOP and this needs to be taken into consideration since the decrease in N₂ fixation may not be as dramatic. What I am suggesting is to put a grain of salt in the last sentence of the abstract to recognize the role of DOP.

Reviewer #3 (Remarks to the Author):

The authors have done a reasonable job of replying to the issues I raised in my first review of the manuscript. As within the original review, the laboratory and field data remain of high quality and the interpretation and conclusions drawn, while still somewhat speculative in places, are reasonable. As within my original review, I still remain a bit sceptical with respect to the offline model calculations, but there is now at least a more complete description within the methods as to how these were carried out. As requested by the editor, I also considered the authors responses to

the original 'reviewer 2'. I am satisfied that the authors have done a good job of replying to these comments, in particular addressing the issues raised around consideration of DOP and around the lack of presentation of a P replete laboratory case as a comparison point. I am therefore still supportive of publication of this work in some form, but if the offline model calculations are to be included I again urge a bit more caution in the emphasis placed on the numerical value of these estimates.

Further related to the authors' response to my original comments regarding the offline model calculations, I note the author's state:

'We consider uncertainties with such model calculations are less compared to a comprehensive modelling simulation and prediction would cause'.

I am not necessarily sure this is the case as it is difficult to quantify the uncertainties in either approach, i.e. the authors don't (and probably cannot) quantify the errors in the calculations they currently undertake and it is obviously not possible to quantify the errors in some hypothetical more mechanistic future study.

Moreover, the authors further state:

'Particularly this will allow the audience who do not have much background in ocean biogeochemistry to better understand the ecological and biogeochemical implications of our study'

I would suggest that the danger here is if the current estimates of potential magnitudes of the effect are over or under stated (which may for example particularly be the case if there are negative / positive feedbacks might occur in the actual system) then a reader might be left with a unrealistic opinion on the importance of the phenomena discussed. For this reason I would still like to see the authors more explicitly state the caveats of the offline modelling in places and be cautious in stating this component of their conclusions. Specific suggestions to this effect follow:

Line 32: given the way the calculations are performed, I think the phrasing here needs to be changed. E.g. 'Estimating the potential implications of our finding using outputs from the CESM predicts that..' would be a more accurate statement.

Line 221: suggest 'To illustrate the potential implications...'

Line 648: 'to illustrate the potential magnitude of impacts'

Manuscript # NCOMMS-21-46616B by Zhang et al.

We thank Reviewers for their additional comments. We respond to these below, indicating further changes to the manuscript.

Reviewer #2

Page 1: I agree with reviewer 1 that normalizing rates of C and N fixation to cellular C and N, respectively, is an important result that should be presented in the paper. While the authors make a strong case as to why the POP-normalized rates are informative in the context of their experimental design, they provide different information and I would recommend presenting (including statistics) and discussing both ways of normalizing. Indeed, knowing that P-limited cells can vary their P content via a myriad of mechanisms, normalizing rates of C and N fixation to P does not tell us how OA affects the ‘specific’ rates which should ideally be C to C and N to N, notwithstanding their respective biases.

We are thankful to the Reviewer for taking time to go through our responses to Reviewer #1 and providing additional comments for improvement. We agree with the Reviewer that C to C and N to N ‘specific’ rates are important results and should be reported. We have now presented and discussed C and N₂ fixation rates normalized to the cellular POC and PON, respectively, in the manuscript (Lines 97-104 and Supplementary Table 2).

Page 2, 3 and 4: The authors answer several questions posed by the reviewer 1 but do not explain how they clarified in the manuscript. This should have been addressed more clearly. I found some of the relevant information in the revised manuscript. However, I think that additional information found in the responses to reviewer 1 should also be found in the manuscript. In particular, the justification as to why the P-limited acidified treatment was not showing lower growth rates as one might expect.

Thanks for the suggestion. We have now included a new section in Supplementary Information to explain growth rates in chemostat cultures, especially why the P-limited acidified treatment was not showing lower growth rates as one might expect (please see Supplementary Note 1). We did not add this information in the main text as we feel that it would distract from the main story we aim to present here.

Page 5: My expertise with modeling is very limited so I would prefer that an expert weigh on this, but thank you for clarifying where to find the changes in the manuscript. L852 of the revised manuscript, you indicate using both DIP and DOP for Trichodesmium’s growth in the model. Could you provide details? Indeed, the labile fraction of DOP being unknown, I am wondering how this is taken into account. If the total concentration of both forms of P are considered in the “limiting factor calculations”, I would expect that Trichodesmium would not be

P limited. Speaking of which, could you explain what this “limiting factor calculations” is?

In our model calculations, phytoplankton growth rates are calculated by multiplying the maximum growth rates by the light- and nutrient-limiting (only the most limiting nutrient) factors. The nutrient limiting factor is calculated based on Michaelis-Menten kinetics. To calculate the P-limitation for *Trichodesmium*, which can use both DIP (phosphate) and DOP for their growth in the Biogeochemical Elemental Cycling (BEC) model, it is formulated as follows:

$$V_i^{\text{PO}_4} = \frac{\text{PO}_4/K_i^{\text{PO}_4}}{1 + \text{PO}_4/K_i^{\text{PO}_4} + \text{DOP}/K_i^{\text{DOP}}} ;$$

$$V_i^{\text{DOP}} = \frac{\text{DOP}/K_i^{\text{DOP}}}{1 + \text{PO}_4/K_i^{\text{PO}_4} + \text{DOP}/K_i^{\text{DOP}}} ; \text{ and}$$

$$V_i^{\text{P}} = V_i^{\text{PO}_4} + V_i^{\text{DOP}}.$$

where V_i^{P} is the limiting factor for P. In this scheme, $K_i^{\text{PO}_4}$ is about 5 times lower than K_i^{DOP} , representing that phosphate is a more preferred P source than DOP in the model.

Similar calculations are conducted for other nutrients (e.g., Fe). The nutrient (either P or Fe) with the smallest V_i is considered as the most limiting nutrient, and the smallest V is used in the growth rate calculations. This is how we determine *Trichodesmium* is P-limited (wherever V_i^{P} is the smallest) in our subsequent calculations. This information has now been added in Methods (Lines 684-692). Although *Trichodesmium* can use both DIP and DOP, their N_2 fixation can still be limited by the overall deficiency of P. For example, it has been shown that in the tropical Atlantic where DIP concentrations were very low, alkaline phosphatase (AP) activity of natural *Trichodesmium* populations was high, and incubation additions of DIP resulted in an increase in N_2 fixation and a loss of AP activity (e.g., Mulholland et al. 2002; Webb et al. 2007).

L95 of the manuscript: the term “dissolved reactive phosphorus” is unusual, why not using “soluble reactive phosphorus”? Are you meaning something different?

We used the method of Yuan et al. (2016) and thus followed the authors to use the term “dissolved reactive phosphorus (DRP)”. DRP has now been replaced by “soluble reactive phosphorus (SRP)” throughout the manuscript (Lines 75, 331, 518).

L202: I suggest replacing the word “significant” since it is a statistical term, and therefore inappropriate here. Perhaps “important” would be a better choice.

Thanks for the suggestion. “significant” has now been replaced by “important” (Line 161).

L219: I suggest removing “indeed”.

Removed as suggested.

L341: The bioavailability of DOP is important, but it is unclear to me how it was considered in the model (see my comment above). It is important because the way the abstract is phrased, one may interpret literally that *Trichodesmium* N₂ fixation will decrease by 38% due to ocean acidification in DIP limited environments. However, we know that *Trichodesmium* can utilize DOP and this needs to be taken into consideration since the decrease in N₂ fixation may not be as dramatic. What I am suggesting is to put a grain of salt in the last sentence of the abstract to recognize the role of DOP.

We refer to our responses to the Reviewer comments above for how DOP was considered in the model calculations. And following the Reviewer’s suggestion, we have now rephrased the last sentence of the abstract as “Estimating the potential implications of this finding, using outputs from the CESM, predicts that acidification and dissolved inorganic and organic P stress could synergistically cause an appreciable decrease in global *Trichodesmium* N₂ fixation by 2100.” (Lines 31-34). In addition, we have emphasized in the concluding paragraph that DOP can be an important P source for *Trichodesmium* and that the effects of OA on DOP utilization and thus N₂ fixation by *Trichodesmium* warrant further investigation (Lines 248-251).

References

- Mulholland, M. R., Flöge, S., Carpenter, E. J. & Capone, D. G. Phosphorus dynamics in cultures and natural populations of *Trichodesmium* spp. *Mar. Ecol. Prog. Ser.* **239**, 45-55 (2002).
- Webb, E. A., Jakuba, R. W., Moffett, J. W. & Dyrman, S. T. Molecular assessment of phosphorus and iron physiology in *Trichodesmium* populations from the western Central and western South Atlantic. *Limnol. Oceanogr.* **52**, 2221-2232 (2007).
- Yuan, Y., Wang, S., Yuan, D. & Ma, J. A simple and cost-effective manual solid phase extraction method for the determination of nanomolar dissolved reactive phosphorus in aqueous samples. *Limnol. Oceanogr. Meth.* **14**, 79-86 (2016)

Reviewer #3

The authors have done a reasonable job of replying to the issues I raised in my first review of the manuscript. As within the original review, the laboratory and field data remain of high quality and the interpretation and conclusions drawn, while still somewhat speculative in places, are reasonable. As within my original review, I still remain a bit sceptical with respect to the offline model calculations, but there is now at least a more complete description within the methods as to how these were carried out. As requested by the editor, I also considered the authors responses to the original 'reviewer 2'. I am satisfied that the authors have done a good job of replying to these comments, in particular addressing the issues raised around consideration of DOP and around the lack of presentation of a P replete laboratory case as a comparison point. I am therefore still supportive of publication of this work in some form, but if the offline model calculations are to be included I again urge a bit more caution in the emphasis placed on the numerical value of these estimates.

We thank the Reviewer for appreciating our efforts in revising the manuscript, and are glad that the Reviewer is satisfied with our responses to issues s/he and Reviewer #2 raised.

We fully agree with the Reviewer that cautions should be taken not to overemphasize the estimates of potential impacts of acidification and P limitation on global *Trichodesmium* N₂ fixation using the offline model calculations. Following the Reviewer's suggestions, we have now (i) rephrased "...reduce global *Trichodesmium* N₂ fixation by 38%..." as "cause an appreciable decreased in global *Trichodesmium* N₂ fixation..." in Abstract (Line 33); (ii) removed "a 38% decrease" in Results and Discussion (Line 232); and (iii) toned down the conclusions and explicitly stated the caveats of the model estimates (Lines 226-232, 234-237, 640-643, 707-710).

Further related to the authors' response to my original comments regarding the offline model calculations, I note the author's state:

'We consider uncertainties with such model calculations are less compared to a comprehensive modelling simulation and prediction would cause'.

I am not necessarily sure this is the case as it is difficult to quantify the uncertainties in either approach, i.e. the authors don't (and probably cannot) quantify the errors in the calculations they currently undertake and it is obviously not possible to quantify the errors in some hypothetical more mechanistic future study.

The Reviewer is correct that we cannot quantify the errors in both our current calculations and more comprehensive model simulations. Our previous response was intended to explain that there will likely be much more unknowns, including both parameters and processes, used in a

more comprehensive model given our current knowledge. It will be even harder to quantify the uncertainties.

Moreover, the authors further state:

‘Particularly this will allow the audience who do not have much background in ocean biogeochemistry to better understand the ecological and biogeochemical implications of our study’

I would suggest that the danger here is if the current estimates of potential magnitudes of the effect are over or under stated (which may for example particularly be the case if there are negative / positive feedbacks might occur in the actual system) then a reader might be left with a unrealistic opinion on the importance of the phenomena discussed. For this reason I would still like to see the authors more explicitly state the caveats of the offline modelling in places and be cautious in stating this component of their conclusions. Specific suggestions to this effect follow:

We thank the Reviewer for these comments and suggestion. Again, we agree with the Reviewer that cautions should be taken in presenting the results of the offline model calculations, and have revised the manuscript following the Reviewer’s comments (Lines 226-232, 234-237, 640-643, 707-710).

Line 32: given the way the calculations are performed, I think the phrasing here needs to be changed. E.g. ‘Estimating the potential implications of our finding using outputs from the CESM predicts that..’ would be a more accurate statement.

Thanks for the suggestion. The sentence has now been rephrased as “Estimating the potential implications of this finding, using outputs from the CESM, predicts that acidification and dissolved inorganic and organic P stress could synergistically cause an appreciable decrease in global *Trichodesmium* N₂ fixation by 2100.” (Lines 31-34).

Line 221: suggest ‘To illustrate the potential implications...’

Rephrased as suggested (Line 226).

Line 648: ‘to illustrate the potential magnitude of impacts’

Rephrased as suggested (Line 658).

REVIEWERS' COMMENTS

Reviewer #3 (Remarks to the Author):

The authors have responded to the comments from myself and other referees and produced a revised manuscript which deals with the majority of points raised well. I note that the authors have now added various caveats associated with the offline model calculations and was glad to see that these now indicate clearly to any reader the potential uncertainties with these estimates. The data remain valuable and the paper well written. As with previous reviews I still think some of the interpretation is speculative in places, but I would not consider this a barrier to publication. I do have one or two further comments that I would like to see the authors consider.

It is now notable that in responding to the comments of reviewer #2 (which themselves related to comments of reviewer #1), the specific rates of C and N fixation are now calculated (see Supplementary Table 2). Given that these experiments were undertaken with a 14h light period, the C specific C fixation for ambient and acidified chemostats were both around 0.25 d⁻¹, which are comparable to the chemostat dilution rates, albeit slightly higher. Although this ~20% difference is probably within the cumulative error of these calculations and/or may indicate some suppression of C fixation during the periods of the day when trichodesmium was actively fixing N₂. However, even assuming 14 hrs of N₂ fixation (which was unlikely given that trichodesmium has to balance C fixation and N fixation during the day, see e.g. Berman-Frank et al. 2001 Science) the N specific N fixation was <0.16 d⁻¹ in both conditions. The authors should comment on this.

The conceptual model on page 18 of the supplementary information is not necessarily the only way that the phosphate limited growth rate relationship may have varied as a function of the pH/pCO₂. I don't think this influences the interpretation of any of the results but would advise the caveat might be included (if indeed the figure is required at all).

Partially related to both the points above, the inclusion of the specific N and C fixation (neither of which significantly vary between the two treatments, which clearly has to be the case as the growth rates are the same), further re-emphasises that the key result is that the lower pH / higher pCO₂ treatment acts to increase cellular POP and thus, as the authors indicate, ultimately the P reduce P use efficiency. The abstract and indeed maybe even the title of the manuscript might be slightly rephrased to clarify this further.

A couple of further minor points:

Line 32: CESM should be defined on first use (apologies I should have spotted this in previous review)

Line 52: '.. limitation by the micronutrient iron...'

Line 200: DRP still used (rather than SRP) here.

Manuscript # NCOMMS-21-46616C by Zhang et al.

Reviewer #3

The authors have responded to the comments from myself and other referees and produced a revised manuscript which deals with the majority of points raised well. I note that the authors have now added various caveats associated with the offline model calculations and was glad to see that these now indicate clearly to any reader the potential uncertainties with these estimates. The data remain valuable and the paper well written. As with previous reviews I still think some of the interpretation is speculative in places, but I would not consider this a barrier to publication. I do have one or two further comments that I would like to see the authors consider.

We are glad that the Reviewer is satisfied with our responses and is supportive of publication of our manuscript. Meanwhile, we are also thankful to the Reviewer for providing further comments for improvement.

It is now notable that in responding to the comments of reviewer #2 (which themselves related to comments of reviewer #1), the specific rates of C and N fixation are now calculated (see Supplementary Table 2). Given that there experiments were undertaken with a 14h light period, the C specific C fixation for ambient and acidified chemostats were both around 0.25 d⁻¹, which are comparable to the chemostat dilution rates, albeit slightly higher. Although this ~20% difference is probably within the cumulative error of these calculations and/or may indicate some suppression of C fixation during the periods of the day when trichodesmium was actively fixing N₂. However, even assuming 14 hrs of N₂ fixation (which was unlikely given that trichodesmium has to balance C fixation and N fixation during the day, see e.g. Berman-Frank et al. 2001 Science) the N specific N fixation was <0.16 d⁻¹ in both conditions. The authors should comment on this.

We thank the Reviewer for this comment. N-specific N₂ fixation rate was calculated by dividing cell number-normalized N₂ fixation rate by PON. The difference between N-specific N₂ fixation rate and chemostat dilution rate, which equals the specific growth rate, is likely caused by the methods we used to measure N₂ fixation rate and PON. In our study, N₂ fixation rates (also called nitrogenase activity) presented in Figure 1 and Supplementary Table S2 were determined by the widely used acetylene reduction assay (Capone 1993), which measured the formation rate of ethylene and then a ratio of 4:1 was applied to convert ethylene production to N₂ fixation, whereas PON was measured by PerkinElmer CHNS/O Analyzer (see Methods). As these two sets of data were measured using two different methods that both may have systematic bias, the calculated N-specific N₂ fixation rate was likely a bit off from the theoretical specific growth rate. Nevertheless, this has no impact on the comparison of the N-specific N₂ fixation rate between the ambient and acidified treatments as reported in the manuscript “Normalizing N₂ and C fixation rates to the cellular PON and POC, respectively, showed that under acidified

conditions the N-specific N₂ fixation rate decreased by 16% ($p = 0.068$, two-tailed paired Student's t-test) and the C-specific C fixation rate was generally not affected ($p = 0.327$, two-tailed paired Student's t-test) (Supplementary Table 2)" (Lines 105-108).

To clarify this, we have now added information on how C-specific C fixation rate and N-specific N₂ fixation rate were obtained in Supplementary Table S2 and also added in the legends of Figure 1 the method we used to measure N₂ fixation rate.

The conceptual model on page 18 of the supplementary information is not necessarily the only way that the phosphate limited growth rate relationship may have varied as a function of the pH/pCO₂. I don't think this influences the interpretation of any of the results but would advise the caveat might be included (if indeed the figure is required at all).

We agreed with the Reviewer that the conceptual model presented in the supplementary information is not necessarily the only possible relationship between growth rates and [PO₄³⁻] under ambient and acidified conditions. We have now added in supplementary information that "It should be noted that the conceptual figure showing here is for illustration purpose, and the two curves (i.e., blue for Ambient and red for Acidified) only represent a possible relationship between growth rate and [PO₄³⁻] under the two treatment conditions."

Partially related to both the points above, the inclusion of the specific N and C fixation (neither of which significantly vary between the two treatments, which clearly has to be the case as the growth rates are the same), further re-emphasises that the key result is that the lower pH / higher pCO₂ treatment acts to increase cellular POP and thus, as the authors indicate, ultimately the P reduce P use efficiency. The abstract and indeed maybe even the title of the manuscript might be slightly rephrased to clarify this further.

Following the Editor and Reviewer's suggestions, we have now revised the Title as

"Phosphate limitation intensifies negative effects of ocean acidification on globally important nitrogen fixing cyanobacterium",

and rephrased the Abstract as (in which we emphasized acidification resulted in increased P demands and decreased P-specific N₂ fixation rate rates)

"Growth of the prominent nitrogen-fixing cyanobacterium *Trichodesmium* is often limited by phosphorus availability in the ocean. How nitrogen fixation of phosphorus-limited *Trichodesmium* may respond to ocean acidification remains poorly understood. Here we use phosphate-limited chemostat experiments to show that acidification enhanced phosphorus demands and decreased phosphorus-specific nitrogen fixation rates in *Trichodesmium*. The increased phosphorus requirements were attributed primarily to elevated cellular polyphosphate

contents, likely for maintaining cytosolic pH homeostasis in response to acidification. Alongside the accumulation of polyphosphate, decreased NADP(H):NAD(H) ratios and impaired chlorophyll synthesis and energy production were observed under acidified conditions. Consequently, the negative effects of acidification were amplified compared to those demonstrated previously under phosphorus sufficiency. Estimating the potential implications of this finding, using outputs from the Community Earth System Model, predicts that acidification and dissolved inorganic and organic phosphorus stress could synergistically cause an appreciable decrease in global *Trichodesmium* nitrogen fixation by 2100.”

A couple of further minor points:

Line 32: CESM should be defined on first use (apologies I should have spotted this in previous review)

CESM has now been defined (Lines 31-32).

Line 52: ‘.. limitation by the micronutrient iron...’

Corrected (Line 52).

Line 200: DRP still used (rather than SRP) here.

Corrected (Line 209), and thanks for pointing this out.

References

Capone, D. G. in: *Handbook of Methods in Aquatic Microbial Ecology* (eds Kemp, P. F., Sherr, B. F., Sherr, E. B. & Cole, J. J.) 621-632 (Lewis Publishers Press, 1993).